# Designing Cu⁰−Cu⁺ dual sites for improved C−H bond fracture towards methanol steam reforming

Hao Meng [1,2], Yusen Yang [1,2] ✉, Tianyao Shen[1], Zhiming Yin[1], Lei Wang[1,2], Wei Liu[1], Pan Yin[1], Zhen Ren[1], Lirong Zheng[3], Jian Zhang [1] ✉, Feng-Shou Xiao [1,4] ✉ & Min Wei [1,2] ✉

Copper-based catalysts serve as the predominant methanol steam reforming material although several fundamental issues remain ambiguous such as the identity of active center and the aspects of reaction mechanism. Herein, we prepare Cu/Cu(Al)O$_x$ catalysts with amorphous alumina-stabilized Cu$_2$O adjoining Cu nanoparticle to provide Cu⁰−Cu⁺ sites. The optimized catalyst exhibits 99.5% CH$_3$OH conversion with a corresponding H$_2$ production rate of 110.8 μmol s$^{-1}$ g$_{cat}$$^{-1}$ with stability over 300 h at 240 °C. A binary function correlation between the CH$_3$OH reaction rate and surface concentrations of Cu⁰ and Cu⁺ is established based on kinetic studies. Intrinsic active sites in the catalyst are investigated with in situ spectroscopy characterization and theoretical calculations. Namely, we find that important oxygen-containing intermediates (CH$_3$O* and HCOO*) adsorb at Cu⁰−Cu⁺ sites with a moderate adsorption strength, which promotes electron transfer from the catalyst to surface species and significantly reduces the reaction barrier of the C−H bond cleavage in CH$_3$O* and HCOO* intermediates.

With increasing resource and environmental challenges, hydrogen is regarded as a potential substitution for fossil energy to a cleaner energy landscape, which has been extensively applied in polymer electrolyte membrane (PEM) fuel cells[1]. However, its practical popularization has been filled with difficulties, especially in respect to hydrogen fuel transport and storage owing to its high risk and the low volume energy density[2,3]. Compared with gaseous H$_2$ storage, in situ production of hydrogen from liquid fuel, such as methanol (CH$_3$OH), not only eliminates safety risks of high-pressure hydrogen storage, but also reduces transportation costs, which provides an alternative solution for fuel cell system application[4,5]. Notably, methanol steam reforming (MSR) serves as one cost-effective way due to the reasonable energy utilization and facile process control, which offers a high

yield of hydrogen at relatively mild reaction conditions[6–8]. MSR as a tandem reaction, which involves complex routes including methanol dehydrogenation, steam reforming and intermediate decomposition, has attracted considerable interest in energy chemistry and heterogeneous catalysis[9,10].

Among various catalysts used in MSR, Cu is an appropriate active ingredient with advantages of cost-effectiveness, satisfactory low-temperature activity, and high H$_2$ selectivity[7,11–13]. Previous studies have focused on tuning Cu crystal structure, doping extra metal constituents (e.g., Ni, Fe, and Co)[14–17], or immobilization on metal oxides supports (e.g., SiO$_2$, Al$_2$O$_3$, TiO$_2$, CeO$_2$, ZnO and ZrO$_2$)[7,12,18–23], so as to boost catalytic performance. Compared with the rapid development of application research, the fundamental scientific insights (such as the

[1]State Key Laboratory of Chemical Resource Engineering, Beijing Advanced Innovation Center for Soft Matter Science and Engineering, Beijing University of Chemical Technology, Beijing 100029, PR China. [2]Quzhou Institute for Innovation in Resource Chemical Engineering, Quzhou 324000, PR China. [3]Institute of High Energy Physics, Chinese Academy of Sciences, Beijing 100049, PR China. [4]Key Lab of Biomass Chemical Engineering of Ministry of Education, College of Chemical and Biological Engineering, Zhejiang University, Hangzhou 310027, PR China. ✉ e-mail: yangyusen@buct.edu.cn; jianzhangbuct@buct.edu.cn; fsxiao@zju.edu.cn; weimin@mail.buct.edu.cn

intrinsic active sites, adsorption behavior and reaction mechanism) of this reaction have not been well solved because of the complex catalyst structure and intricate reaction processes, which could lead to uncertain and even contradictory conclusions. Due to the rich redox properties of copper, various Cu species ($Cu^0$, $Cu^{\delta+}/Cu^+$) normally coexists under actual reaction conditions[23–26]; moreover, the effects of electron rearrangement in alloy, strong metal-support interaction (SMSI) and oxygen vacancy induction make this issue rather complicated[18–22,27,28], which is prone to cause ambiguous relationship between catalytic performance and microscopic structure. Therefore, a detailed study based on spatially and temporally-resolved *operando* characterization techniques, kinetic investigations as well as theoretical calculations is imperative to shed light on the intrinsic active sites, structure-activity correlation, and reaction mechanism. An in-depth understanding of these fundamental issues would not only provide rational criteria for the structure design of heterogeneous catalysts, but also promote further progress of applied technology.

Inspired by the above though, herein, a series of $y$Cu/Cu(Al)O$_x$ samples ($y$ denotes the mass ratio of Cu/Al) with tunable synergistic $Cu^0$–$Cu^+$ sites were prepared through a co-precipitation method followed by subsequent reduction treatment, which were characterized by amorphous alumina-stabilized $Cu_2O$ adjoining Cu nanoparticle. Typically, the 4.25Cu/Cu(Al)O$_x$ sample exhibits the optimal catalytic performance with $a > 99\%$ methanol conversion and a high $H_2$ production rate (110.8 $\mu$mol s$^{-1}$ g$_{cat}^{-1}$), which is preponderant to the previously reported Cu-based catalysts for MSR. The kinetic studies, in situ FT-IR spectroscopy and mass spectrometry analysis substantiate that the MSR reaction follows three main processes: dehydrogenation of $CH_3OH$, hydrolysis of $HCOOCH_3$ and decomposition of $HCOO^*$, where the cleavage of C−H bonds in $CH_3O^*$ and $HCOO^*$ intermediates is the rate-determining step. As revealed by STEM-EELS, isotope dynamics measurements, in situ FT-IR spectra, in situ XAFS spectra and DFT calculation, the synergistic effect between $Cu^0$ and $Cu^+$ species derived from Cu-Cu(Al)O$_x$ boundary plays a crucial role: the oxygen-containing intermediates ($CH_3O^*$ and $HCOO^*$) undergo activation adsorption at $Cu^0$–$Cu^+$ interfacial sites via oxygen-end bridge with a moderate strength. This unique adsorption configuration promotes electron transfer from catalyst surface to reaction intermediates and significantly reduces the energy barrier of C−H bonds cleavage (rate-determining step) in $CH_3O^*$ and $HCOO^*$ intermediates, accounting for the exceptional activity of 4.25Cu/Cu(Al)O$_x$ catalyst.

## Results
### Structural characterizations
The $y$Cu/Cu(Al)O$_x$ samples were prepared via a co-precipitation method, followed by the subsequent roasting and reduction processes (Supplementary Figs. 1 and 2). As shown in Supplementary Fig. 3, the XRD patterns of catalyst precursors ($y$CuAlO$_x$ samples) show characteristic diffraction peaks indexed to a CuO phase (PDF#48-1548). Then, the reduction of $y$CuAlO$_x$ in a $H_2$ atmosphere results in the formation of $y$Cu/Cu(Al)O$_x$ samples, in which both Cu (PDF#04-0836) and $Cu_2O$ (PDF#78-2076) phases are observed (Fig. 1a). The absence of $Al_2O_3$ reflections indicates an amorphous phase. For the control sample 4.20Cu/Al$_2$O$_3$, the reflections ascribed to Cu and $\gamma$-Al$_2$O$_3$ phases are present, without $Cu_2O$ species. When the Cu/Al ratio increases from 0.95 to 7.18, the relative peak intensity of metallic Cu enhances gradually whilst that of $Cu_2O$ declines accordingly, which indicates that the decrease of Al content favors the transformation from $Cu^+$ to $Cu^0$ during $H_2$ reduction. This is in good agreement with XRD Rietveld refinement analysis (Supplementary Figs. 4–10). Notably, compared with the standard $Cu_2O(111)$ lattice plane (36.4°), the $Cu_2O(111)$ reflection in these samples shifts to a higher diffraction angle (36.8°, Fig. 1b) and is located between $Cu_2O(111)$ and $CuAlO_2(101)$ (PDF#40-1037), which indicates that partial $Cu^+$ atoms can be stabilized by amorphous $Al_2O_3$ at the interfacial sites to form a $CuAlO_2$-like structure.

Furthermore, $H_2$-TPR measurements on these $y$CuAlO$_x$ samples were carried out to investigate the structural evolution from catalyst precursor to $y$Cu/Cu(Al)O$_x$. As shown in Supplementary Fig. 11, the $H_2$ consumption peaks at lower temperature ($\alpha$, 150−200 °C) and higher temperature ($\beta$, 200−300 °C) are attributed to the reduction of copper oxide species that weakly and strongly interacts with alumina, respectively. As the Cu/Al ratio increases from 0.95 to 7.18, the intensity of $\alpha$ reduction peak increases whilst $\beta$ reduction peak decreases gradually, which is consistent with the variation tendency of Cu and $Cu_2O$ species in XRD results (Fig. 1a).

The surface chemical states of $y$Cu/Cu(Al)O$_x$ samples were measured by *quasi*-in situ XPS and AES spectra (Supplementary Figs. 12–14). The Cu $2p_{3/2}$ XPS spectra confirm the co-existence of $Cu^+$/$Cu^0$ species (932.2–932.5 eV) with less than 20% $Cu^{2+}$ (934.7–934.9 eV) for all these samples (Supplementary Fig. 12)[20,23]. Auger Cu LMM spectra further verify that the surface $Cu^+$/$Cu^0$ ratio decreases successively as the Cu/Al ratio increases from 0.95 to 7.18 (Fig. 1c and Supplementary Table 1)[29,30]. In contrast, the control sample 4.20Cu/Al$_2$O$_3$ shows the minimal value of $Cu^+$/$Cu^0$ ratio. Furthermore, in situ CO-DRIFTS was performed with CO as the probe molecule to study the surface chemical valence of Cu species. As shown in Fig. 1d, the absorption bands at ($\alpha$) 2105−2107 cm$^{-1}$ and ($\beta$) 2094−2096 cm$^{-1}$ are ascribed to CO bound to $Cu^+$ and $Cu^0$ species, respectively[22,31], in which the variation of surface $Cu^+$/$Cu^0$ ratio ($\alpha$/$\beta$) follows a similar tendency to that of *quasi*-in situ Cu LMM spectra.

The electronic structure and coordination state of copper species were studied in detail through X-ray absorption spectroscopy (XAS). As shown in the normalized Cu K-edge XANES spectra (Fig. 1e), the white line peaks of $y$Cu/Cu(Al)O$_x$ and 4.20Cu/Al$_2$O$_3$ are located between Cu foil and $Cu_2O$ standard, and the latter sample gives the lowest oxidation state (rather close to Cu foil). For the $y$Cu/Cu(Al)O$_x$ samples, the intensity of white line peaks decreases gradually from 0.95Cu/Cu(Al)O$_x$ to 7.18Cu/Cu(Al)O$_x$, indicative of a decline in average valence state of Cu species (Cu$_{AVS}$). This is in accordance with the results from linear combination fitting (LCF) analysis (Supplementary Fig. 15), where the average valence state of Cu (Cu$_{AVS}$) decreases gradually from +0.86 to +0.25 from 0.95Cu/Cu(Al)O$_x$ to 7.18Cu/Cu(Al)O$_x$ sample. In contrast, the control sample Cu/Al$_2$O$_3$ displays the lowest Cu$_{AVS}$ (+0.12). The corresponding Cu K-edge extended X-ray absorption fine structure (EXAFS) spectra from Fourier transform are shown in Fig. 1f and Supplementary Fig. 16, where the peaks at 1.47 and 2.25 Å (no calibration) are assigned to Cu−O and Cu−Cu bonds from the first shell of $Cu_2O$ and Cu foil, respectively[26,28]. For the $y$Cu/Cu(Al)O$_x$ samples, the Cu−O bond length is longer than that in $Cu_2O$ sample. Based on the fitting results and wavelet transform (Supplementary Fig. 17 and Supplementary Table 2), the longer bond length of Cu−O in $y$Cu/Cu(Al)O$_x$ samples ($1.84 \pm 0.01$ Å) relative to $Cu_2O$ standard ($1.81 \pm 0.01$ Å) indicates a distorted tetrahedral structure due to the partial substitution of Cu by Al, which is possibly related to the formation of unique Cu−O−Al geometric coordination ($CuAlO_2$-like structure). This is consistent with the DFT calculation results (Fig. 1g–i), in which the $Cu^+$−O bond length in $Cu(111)/CuAlO_2(101)$ (1.90 Å) is significantly longer than that in $Cu_2O(111)$ (1.83 Å) and $Cu(111)/Cu_2O(111)$ (1.83 Å). In addition, the coordination number of Cu−Cu bond in $y$Cu/Cu(Al)O$_x$ increases from 5.1 to 8.2 whilst that of Cu−O bond declines from 1.7 to 0.5 along with the increment of Cu/Al ratio (Fig. 1f, Supplementary Fig. 17 and Supplementary Table 2). We further correlated the coordination number of Cu−O and Cu−Cu bonds with the fraction of $Cu_2O$/Cu in these samples. Based on the XRD Rietveld refinement, *quasi*-in situ Cu LMM, in situ CO-DRIFTS, XAFS-LCF, and EXAFS-Fit analysis results, a negative correlation between $Cu^+$/$Cu^0$ ratio and Cu/Al ratio is established, demonstrating the significant role of amorphous alumina in stabilizing $Cu^+$ species (Supplementary Fig. 18).

In addition, we changed the calcination temperature of 4.25CuAlO$_x$ precursor within 500−800 °C to regulate the doping

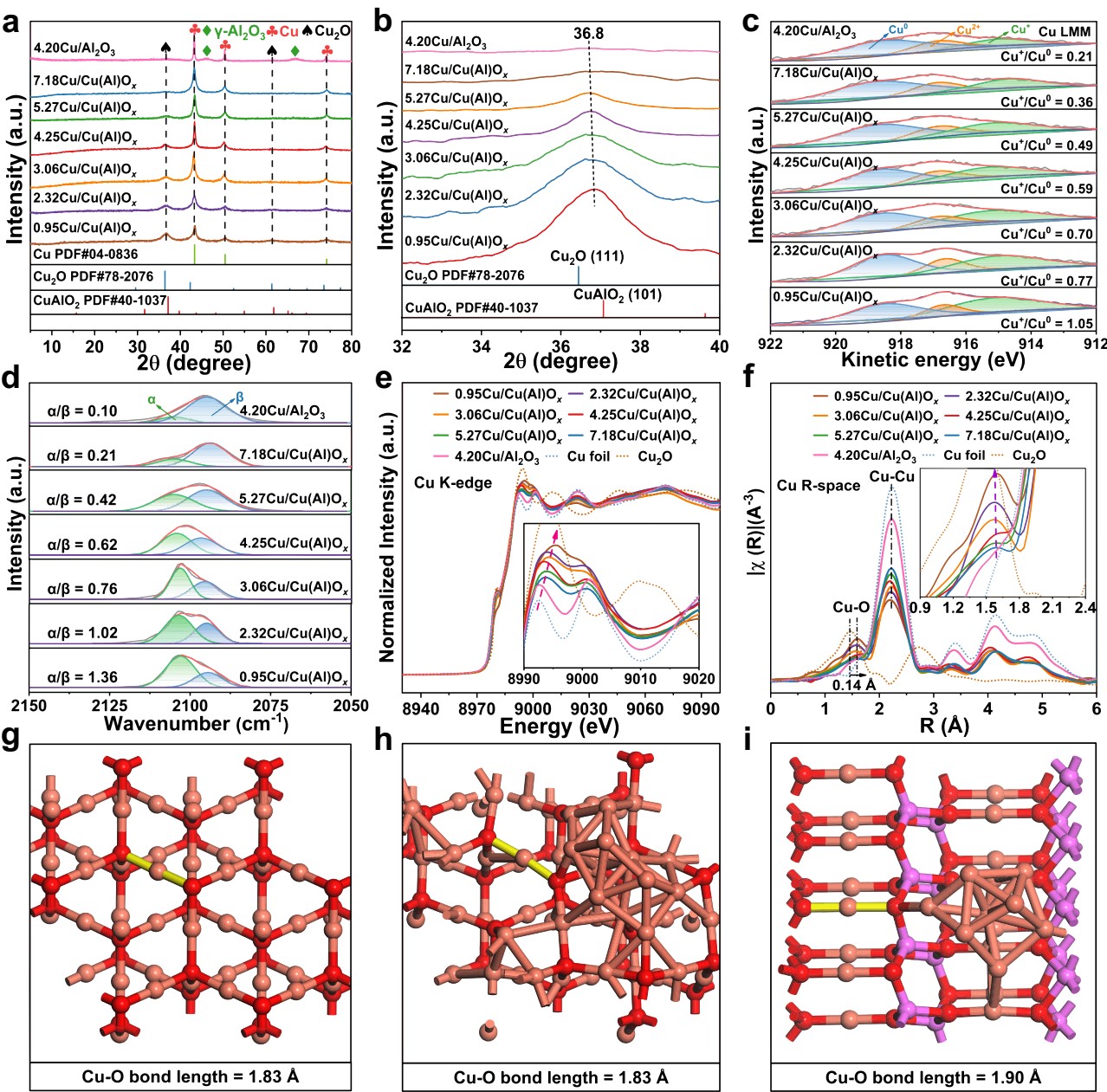

**Fig. 1 | Fine structure characterizations of various samples. a, b** XRD patterns, **c** *quasi*-in situ Cu LMM AES spectra, **d** CO-DRIFT spectra, **e** Cu K-edge XANES and **f** Cu K-edge EXAFS spectra of *y*Cu/Cu(Al)O$_x$ and control samples. Cu−O bond length in **g** Cu$_2$O(111), **h** Cu(111)/Cu$_2$O(111) and **i** Cu(111)/CuAlO$_2$(101) based on DFT calculations (red: O; orange: Cu; purple: Al).

degree of amorphous alumina on Cu$_2$O. As shown in the H$_2$-TPR curves (Supplementary Fig. 19a), the reduction peak moves towards higher temperature with the increase of precursor roasting temperature, signifying an enhanced Cu-Al$_2$O$_3$ interaction. After the subsequent hydrogen activation treatment, XRD patterns show that from 4.25Cu/Cu(Al)O$_x$−500 to 4.25Cu/Cu(Al)O$_x$−800 sample (Supplementary Fig. 19b, c), the Cu$_2$O(111) reflection shifts to higher 2θ direction accompanied with an increased peak intensity, which indicates a decreased Cu$_2$O cell volume due to the doping of Al atoms with smaller radius. Moreover, XPS (Supplementary Fig. 19d) and XANES spectra (Supplementary Fig. 19e) demonstrate a gradual increase of Cu$^+$/Cu$^0$ ratio from 4.25Cu/Cu(Al)O$_x$−500 to 4.25Cu/Cu(Al)O$_x$−800. Meanwhile, an enhanced proportion of Cu−O−Al geometric coordination is further verified through fitting the Cu−O and Cu−O−Al scattering paths based on Cu K-edge EXAFS spectra with Cu$_2$O and CuAlO$_2$ as standard

samples (Supplementary Figs. 19f, 20 and Supplementary Table 3). The average bond length of Cu−O increases from 1.84 ± 0.01 (4.25Cu/ Cu(Al)O$_x$−500) to 1.89 ± 0.02 Å (4.25Cu/Cu(Al)O$_x$−800) in sequence, accompanied with a gradual increase in the coordination number of Cu−Al bond from the second shell (bond length: ~3.17 Å), which indicates the formation of Cu−O−Al geometric coordination. In addition, we built 7 modified Cu$_2$O models in which Cu$^+$ is substituted by Al$^{3+}$ with 7 different replacement percentages (Supplementary Fig. 21a−n). The calculated average bond length of Cu−O enhances from 1.83 to 1.87 Å as the Al/Cu ratio increases from 0 to 7.7% (Supplementary Fig. 21o), in agreement with the experiment results.

TEM images of these *y*Cu/Cu(Al)O$_x$ samples show a uniform dispersion of Cu nanoparticles on substrate (Supplementary Fig. 22); and the average particle size increases from 7.0 to 9.3 nm along with the increment of Cu/Al ratio from 0.95 to 7.18. As shown in the HR-TEM

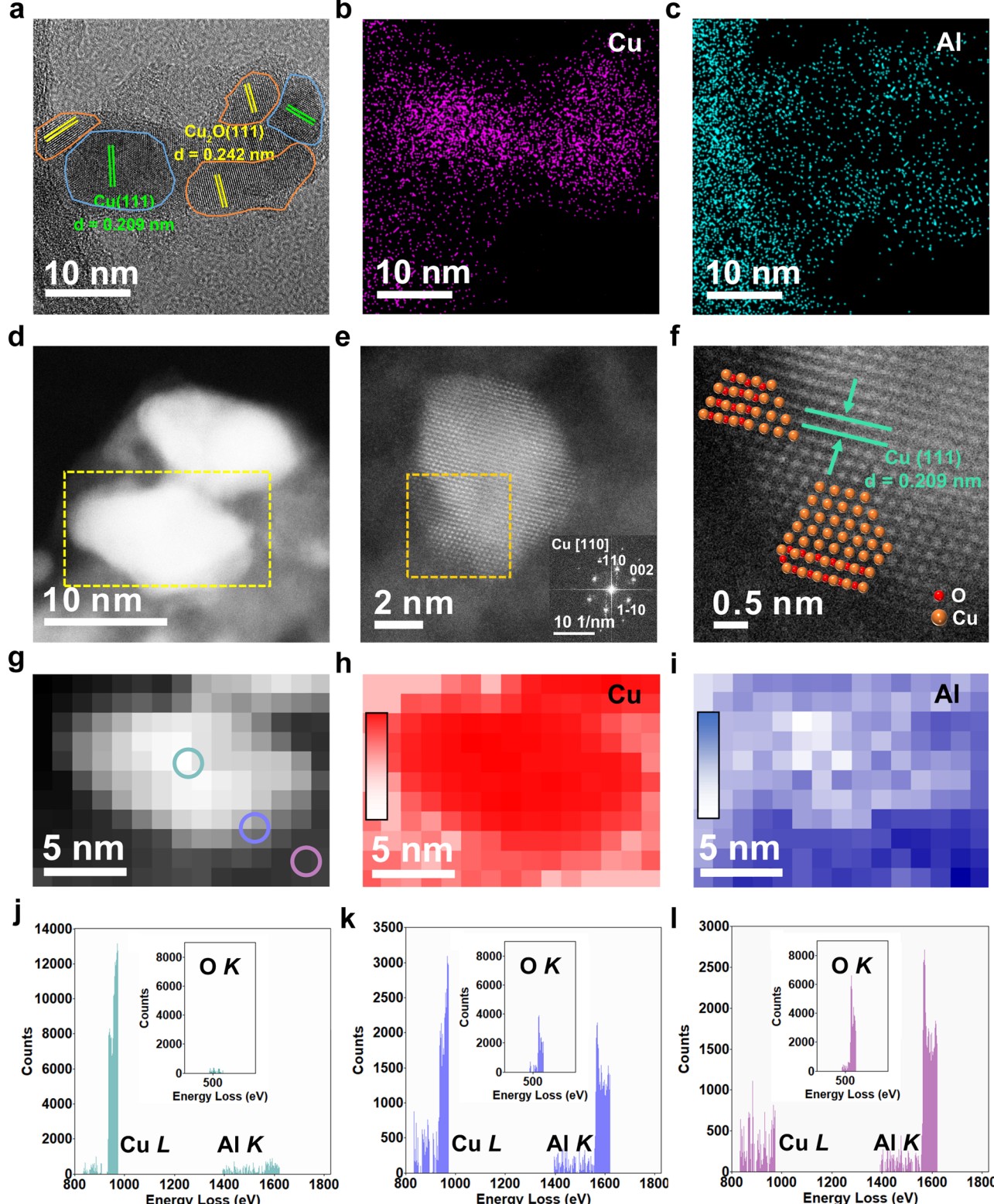

**Fig. 2 | Microscopy characterizations of the 4.25Cu/Cu(Al)Ox sample. a** High-magnification TEM, **b**, **c** EDS mapping images, **d**–**f** High-magnification STEM-HAADF images, **g** EELS mapping image of 4.25Cu/Cu(Al)O$_x$ sample in yellow box in (**d**). Elemental areal density of **h** Cu and **i** Al in **g** (color bars from bottom to top indicates increased intensity). **j**–**l** EELS signals of Cu $L$, Al $K$, and O $K$ at different tagged sites in (**g**).

images (Fig. 2a and Supplementary Fig. 23), two clear crystalline phases are identified for these $y$Cu/Cu(Al)O$_x$ samples: the lattice fringe of 0.209 nm corresponds to the Cu(111) plane and that of 0.242 nm around Cu species is due to the Cu$_2$O(111) plane from face-centered cubic packing. Combining with the EDS mapping results (Fig. 2b, c and

Supplementary Fig. 24), stabilized Cu or Cu$_2$O phase on the Al$_2$O$_3$ support is verified. In contrast, the control sample 4.20Cu/Al$_2$O$_3$ shows an average Cu particle size of 7.1 nm (Supplementary Fig. 25a), with the presence of individual Cu(111) plane (Supplementary Fig. 25b). To clearly reveal the microstructure, STEM and corresponding EELS

mapping for the typical sample 4.25Cu/Cu(Al)O$_x$ were measured. As shown in STEM-HAADF (Fig. 2d–f and Supplementary Fig. 26) and STEM-BF (Supplementary Fig. 27) images, this sample is constituted by Cu nanoparticle and its surrounding O-terminal Cu species, in agreement with the XRD and XAFS results. Furthermore, three sites are selected (Fig. 2g) to carry out EELS element analysis for identifying the fine structure. At the surface site on copper particle (green tagged), Cu$^0$ species displays overwhelming superiority with a very small amount of Al and O (Fig. 2j). At the boundary site between Cu particle and support (blue tagged), a simultaneous presence of Cu $L$, Al $K$, and O $K$ is observed (Fig. 2k), corresponding to the Cu$^+$ species stabilized by amorphous alumina. At the site away from Cu particle (pink tagged), the EELS signal of Cu shrinks accompanied with enhanced signals of Al and O (Fig. 2l). These results are in accordance with the areal density of Cu (Fig. 2h) and Al (Fig. 2i) in the 4.25Cu/Cu(Al)O$_x$ sample, where the

relative content of copper decreases while that of aluminum increases gradually from the Cu particle surface to the Cu–Al$_2$O$_3$ interface. As shown in STEM and EELS, the existence of Cu$_2$O phase (Fig. 2d–l) is related to the stabilizing effect of amorphous alumina support on Cu$^+$, which suppresses its further reduction. Thus, the 4.25Cu/Cu(Al)O$_x$ sample is featured by aluminum-stabilized Cu$^+$ adjacent to Cu$^0$ nanoparticle immobilized on Al$_2$O$_3$ support, whose schematic structure diagram is shown in Supplementary Fig. 28 (Supplementary Note 1).

## Catalytic performance and kinetic studies

Catalytic performances of $y$Cu/Cu(Al)O$_x$ samples were evaluated towards ESR reaction in a fix-bed reactor at 180–270 °C with a H$_2$O/CH$_3$OH molar ratio of 2 in feed. As an endothermic reaction, both CH$_3$OH conversion and H$_2$ production rate increase upon elevating reaction temperature (Fig. 3a, b); while CO$_2$ selectivity maintains

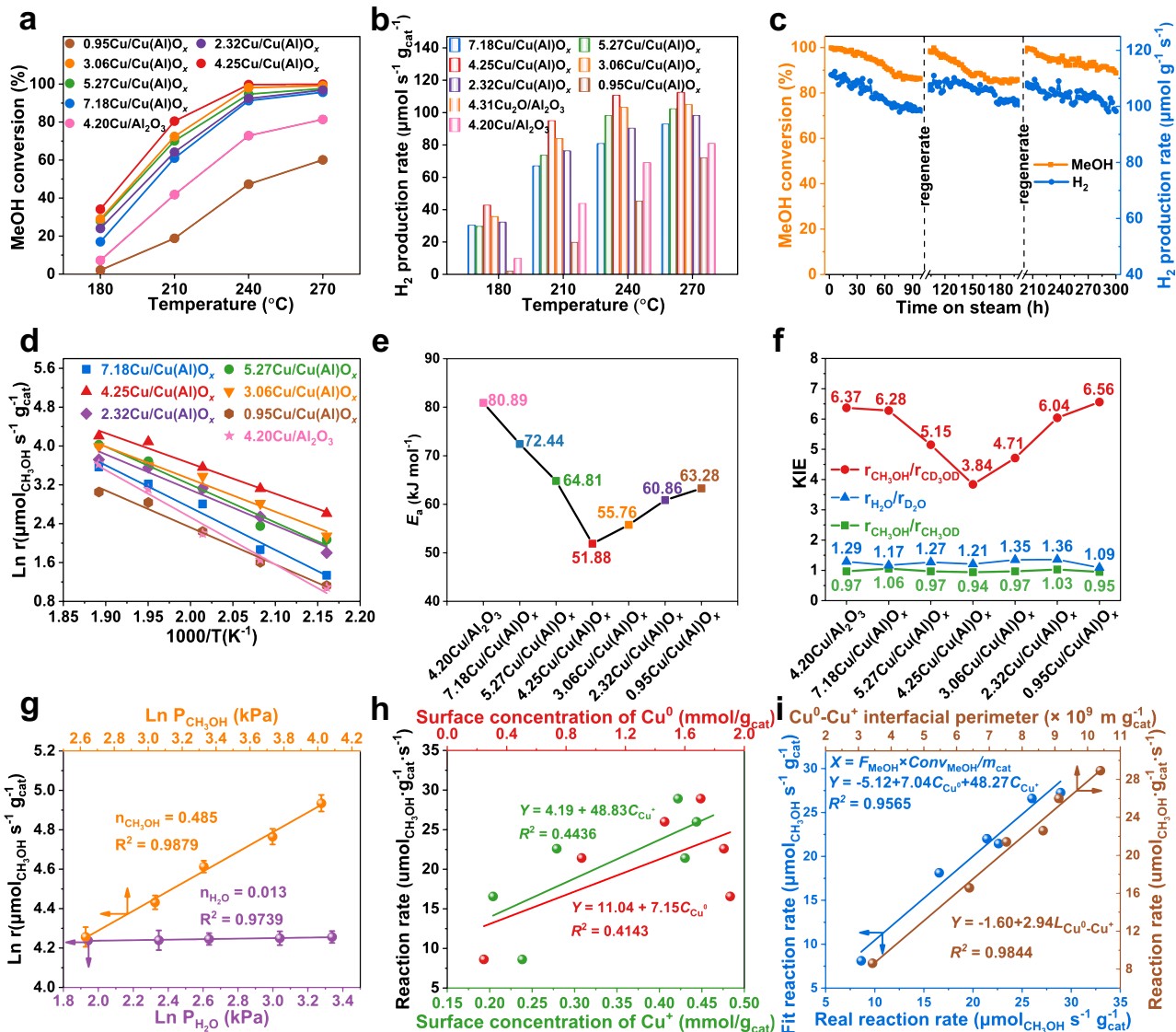

**Fig. 3 | Catalytic performance towards MSR and kinetic studies on various samples. a** Methanol conversion and **b** H$_2$ production rate over various samples within 180–270 °C (Reaction conditions: catalyst (0.25 g) + SiO$_2$ (2.50 g); liquid feed of S/C = 2 at 0.040 mL min$^{-1}$; He carrier at 50.0 mL min$^{-1}$; time on stream: 1.0 h). **c** Methanol conversion and H$_2$ production rate vs. reaction time on stream of 4.25Cu/Cu(Al)O$_x$ catalyst at 240 °C. **d** Kinetic studies on Arrhenius plots, **e** activation energy ($E_a$) and **f** KIE values for MSR reaction over various samples (Reaction conditions: catalyst (0.01–0.10 g) + SiO$_2$ (0.1–1.0 g); liquid feed of S/C = 2

at 0.040–0.080 mL min$^{-1}$; He carrier at 50.0 mL min$^{-1}$; time on stream: 0.5 h; methanol conversion less than 20%). **g** Reaction orders of CH$_3$OH and H$_2$O over 4.25Cu/Cu(Al)O$_x$ catalyst at 240 °C (error bar comes from the uncertainty obtained from three parallel experiments). **h** Correlation between methanol reaction rate and surface concentration of individual Cu$^0$ ($C_{Cu}0$) or Cu$^+$ ($C_{Cu}+$). **i** Linear fitting results of methanol reaction rate as a function of both Cu$^0$ ($C_{Cu}0$) and Cu$^+$ ($C_{Cu}+$) as well as Cu$^0$–Cu$^+$ interfacial perimeter.

above 98% for all these samples (Supplementary Fig. 29). Interestingly, a volcanic tendency of catalytic activity is found from 0.95Cu/Cu(Al)$O_x$ to 7.18Cu/Cu(Al)$O_x$; and the 4.25Cu/Cu(Al)$O_x$ catalyst exhibits the highest CH$_3$OH conversion (99.5%) as well as H$_2$ production rate (110.8 µmol s$^{-1}$ g$_{cat}$$^{-1}$) at 240 °C, which is preponderant to the reported copper-based catalysts for MSR (Supplementary Table 4) and even exceeds most precious metal catalysts. In addition, the studies on reduction temperature from 170 to 300 °C (Supplementary Fig. 30) show that the 4.25Cu/Cu(Al)$O_x$ sample reduced at 220 °C with an appropriate proportion of Cu$^+$ species (Supplementary Fig. 31) displays the highest catalytic activity. Furthermore, the stability test of 4.25Cu/Cu(Al)$O_x$ was also examined (Fig. 3c). Although the CH$_3$OH conversion and H$_2$ production rate show somewhat decrease after a 100 h time-on-stream test (from 99.5% and 110.8 µmol s$^{-1}$ g$_{cat}$$^{-1}$ to 86.3% and 99.4 µmol s$^{-1}$ g$_{cat}$$^{-1}$), the catalytic performance can recover to its original level after a regeneration process (air oxidation at 300 °C for 1 h followed by a reduction in 25% H$_2$/N$_2$ at 220 °C for 1 h). No significant change in catalyst structure, Cu particle size, and chemical valence is observed for the used catalyst after three-cycle tests (300 h), in comparison with the fresh one (Supplementary Figs. 32–34), which indicates that the decrease of catalytic activity is not associated with the variation in physicochemical properties. As proved by IR spectroscopy (Supplementary Fig. 35), the deactivation of 4.25Cu/Cu(Al)$O_x$ catalyst primarily arises from carbon deposition (the band between 2800 and 3000 cm$^{-1}$ is assigned to C−H species)[24], and the carbonaceous species can be facilely removed via a regeneration process.

In addition, the activation energy ($E_a$) and kinetic isotope effect (KIE) were then studied over these $y$Cu/Cu(Al)$O_x$ samples under dynamics test conditions. According to the Arrhenius equation (Fig. 3d), the control sample 4.20Cu/Al$_2$O$_3$ gives the largest activation energy of 80.89 kJ mol$^{-1}$. In the cases of $y$Cu/Cu(Al)$O_x$ samples, the activation energy declines first and then increases from 0.95Cu/Cu(Al)$O_x$ to 7.18Cu/Cu(Al)$O_x$, and the lowest value of 51.88 kJ mol$^{-1}$ is present in the 4.25Cu/Cu(Al)$O_x$ catalyst (Fig. 3e), in accordance with the highest catalytic activity. Similar volcanic curves were obtained by correlating the methanol reaction rate with the surface Cu$^+$/Cu$^0$ ratio (Supplementary Fig. 36). To acquire clear-cut kinetics information, the KIE values of D$_2$O, CH$_3$OD, and CD$_3$OD were further tested (Fig. 3f). The k$_H$/k$_D$ values of D$_2$O and CH$_3$OD are estimated to be 1.09−1.36 and 0.95−1.06, respectively; in contrast, the k$_H$/k$_D$ value of CD$_3$OD is located between 3.84 and 6.56, several times larger than the former two cases, indicating that the breakage of C−H bonds is the rate-determining step during the MSR reaction. Importantly, a similar volcanic-type profile for the catalytic activity as a function of k$_H$/k$_D$ value of CD$_3$OD is found from 0.95Cu/Cu(Al)$O_x$ to 7.18Cu/Cu(Al)$O_x$ (red line); whilst the KIE values of CH$_3$OD (green line) and D$_2$O (blue line) give no significant difference on these samples. In addition, we measured the reaction order of CH$_3$OH and H$_2$O over the 4.25Cu/Cu(Al)$O_x$ catalyst (Fig. 3g). The reaction rate of methanol displays a positive relationship with the CH$_3$OH partial pressure, but does not show obvious correlation with H$_2$O partial pressure. Through data fitting, the reaction orders of CH$_3$OH and H$_2$O are determined to be 0.485 and 0.013, respectively, which indicates that the cleavage of C−H bond in methanol is crucial whilst the H$_2$O activation is not involved in the rate-determining step of MSR reaction. The results above demonstrate that the 4.25Cu/Cu(Al)$O_x$ catalyst promotes the breakage of C−H bond, which is responsible for its excellent catalytic activity.

Based on the structural characterizations and catalytic evaluations, the Cu$^0$−Cu$^+$ synergistic catalysis in $y$Cu/Cu(Al)$O_x$ samples would play a key role. Thus, we quantified the surface concentration of Cu$^0$ ($C_{Cu}$0) and Cu$^+$ ($C_{Cu^+}$) species as well as interfacial perimeters of Cu$^0$−Cu$^+$ ($L_{Cu^0−Cu^+}$) of all these samples via N$_2$O titration (Supplementary Fig. 37) and CO-TPD (Supplementary Fig. 38). No obvious correlation between normalized CH$_3$OH reaction rate and individual

$C_{Cu}$0 or $C_{Cu^+}$ is found (Fig. 3h). Interestingly, if we correlate the CH$_3$OH reaction rate with $C_{Cu}$0 and $C_{Cu^+}$ simultaneously, a binary function relation of rate equation is obtained (Fig. 3i): $Y = -5.12 + 7.04 C_{Cu}0 + 48.27 C_{Cu^+}$. Furthermore, a linear relationship between methanol reaction rate and $L_{Cu^0−Cu^+}$ is obtained (Fig. 3i). It is thus concluded that catalytic activity depends on the synergistic catalysis of Cu$^0$ and Cu$^+$ rather than a single active site, and the Cu$^0$−Cu$^+$ interfacial sites are imperative for boosting the rate-determining step in MSR reaction (C−H bond cleavage).

## Catalytic reaction mechanism of MSR

For the MSR reaction, three possible pathways have been discussed in the previous studies: (1) CH$_3$OH decomposition to CO and H$_2$, followed by CO steam reforming to produce CO$_2$ and H$_2$ (denoted as CO* route)[4–6,32]; (2) one step oxidization of CH$_3$OH to HCOO* by hydroxyl group or reactive oxygen species from H$_2$O dissociation, followed by HCOO* decomposition to produce CO$_2$ and H$_2$ (denoted as HCOO* route)[7,33–35]; (3) CH$_3$OH dehydrogenation to HCOOCH$_3$, which is then hydrolyzed to HCOO*, followed by further decomposition to produce CO$_2$ and H$_2$ (denoted as HCOOCH$_3$* route)[9,10,36,37]. Nonetheless, owing to the complexity of MSR reaction and the diversity of catalyst structure, studies on catalytic reaction pathways over copper catalysts are controversial.

To elucidate the MSR reaction route over $y$Cu/Cu(Al)$O_x$ catalysts in this work, the *operando* pulse experiments equipped with mass spectrometer detector were carried out (Fig. 4 and Supplementary Fig. 39). As shown in Fig. 4a, the signals of reaction intermediates (CH$_3$O*, HCHO, HCOOCH$_3$ and HCOO*) and reaction products (H$_2$ and CO$_2$) are captured after the introduction of CH$_3$OH/He on 4.25Cu/Cu(Al)$O_x$ at 240 °C (Fig. 4c, red). In contrast, when CH$_3$OH + H$_2$O is co-introduced (Fig. 4b), the relative intensity of HCOOCH$_3$ decreases accompanied with the increase of HCOO* signal (Fig. 4c, blue). This indicates that the co-introduction of H$_2$O greatly promotes the hydrolysis of HCOOCH$_3$. The same results are also found on 0.95Cu/Cu(Al)$O_x$ and 7.18Cu/Cu(Al)$O_x$ samples, but the higher relative intensities of HCOO* and CH$_3$O* suggest a lower reaction rate (Supplementary Fig. 39).

Furthermore, we performed the pulse experiments of HCOOCH$_3$ and HCOOCH$_3$ + H$_2$O over 4.25Cu/Cu(Al)$O_x$ catalyst at different reaction temperatures (Fig. 4d). In the former case, as the temperature increases from 150 to 240 °C, the signals of HCOOCH$_3$ decline accompanied with the rise of H$_2$, CO$_2$, CH$_3$O* and HCOO* signals due to the dissociation of HCOOCH$_3$. In the latter case, the presence of H$_2$O promotes the conversion of HCOOCH$_3$, with significantly weakened HCOOCH$_3$ signal but enhanced CH$_3$O* and HCOO* signals. Therefore, the whole MSR reaction mechanism over 4.25Cu/Cu(Al)$O_x$ catalyst is proposed in Fig. 4e: CH$_3$OH firstly undergoes dehydrogenation to form CH$_3$O* and HCHO species; then HCHO experiences dimerization or reacts with CH$_3$O* to generate HCOOCH$_3$; subsequently, HCOOCH$_3$ hydrolyzes to form HCOOH and CH$_3$O*, and CH$_3$O* re-participates in the catalytic cycle; finally, the decomposition of HCOOH occurs to produce CO$_2$ and H$_2$. In addition, according to the MS spectra results (Fig. 4c, Supplementary Fig. 39, and Supplementary Note 2), the signals of CH$_3$O* and HCOO* are much stronger than those of HCHO and HCOOCH$_3$ during the MSR reaction, indicating that the conversion of CH$_3$O* and HCOO* is a kinetically slower process. According to the kinetic studies (Fig. 3f, g), the fracture of C−H bonds in CH$_3$O* and HCOO* intermediates is proved as the rate-determining step; moreover, water molecule promotes the decomposition of HCOOCH$_3$ but does not participate directly in the cleavage of C−H bonds.

In situ FT-IR measurements were carried out to study the microscopic reaction mechanism of MSR. As shown in Fig. 5a, b, when introducing CH$_3$OH to the 4.25Cu/Cu(Al)$O_x$ catalyst at 240 °C (red lines), signals including dissociative adsorption of CH$_3$OH (1032 and 1056 cm$^{-1}$), bending vibration of HCOO* (1327, 1351 and 1583 cm$^{-1}$) and CH$_3$O* species (1458 and 1477 cm$^{-1}$), stretching vibration of C−H bond

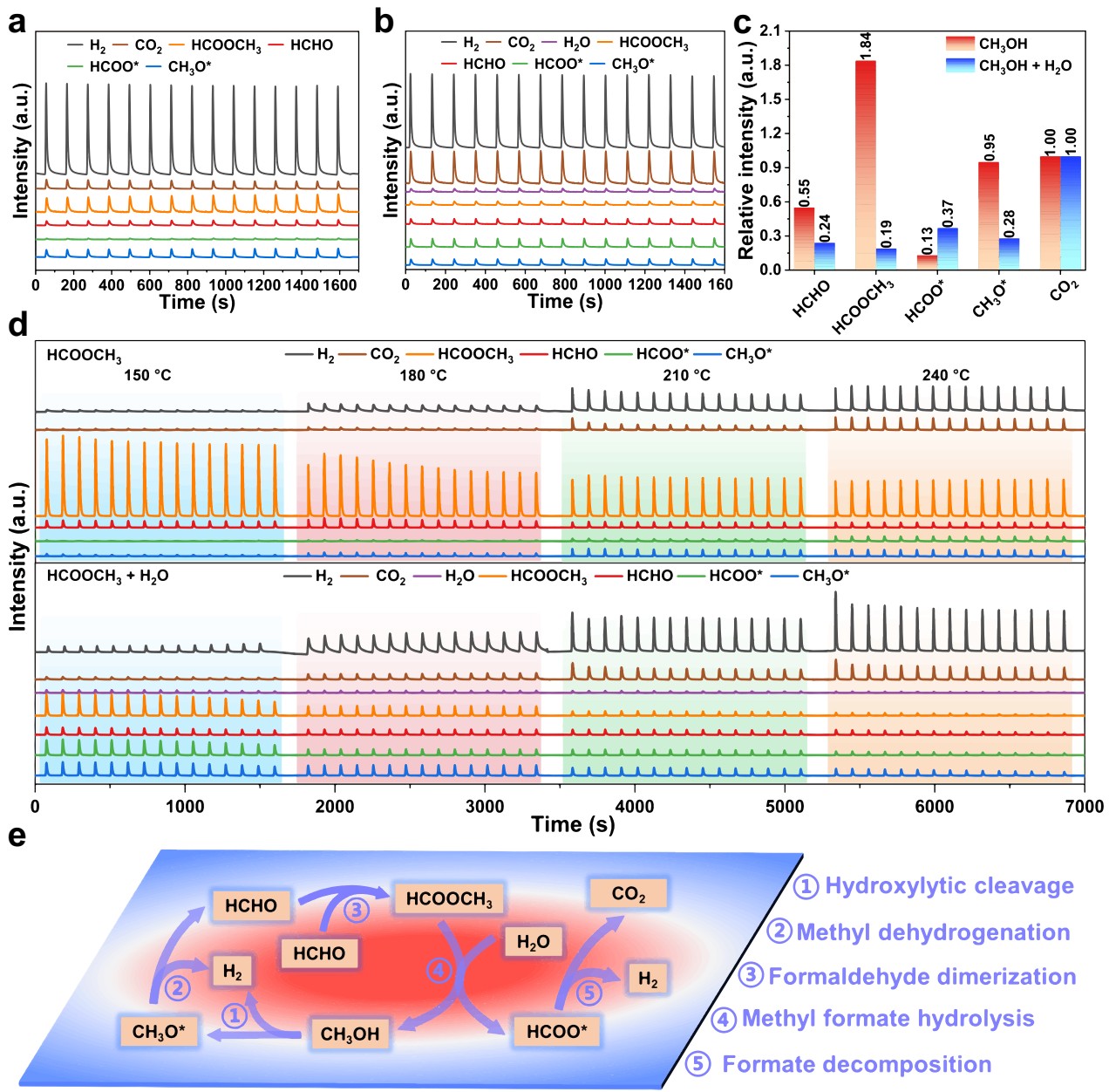

**Fig. 4 | Operando pulse experiments and reaction mechanism of MSR.** MS signals for the pulse experiments of **a** methanol and **b** methanol-water (1:2) over 4.25Cu/Cu(Al)O$_x$ at 240 °C, respectively. **c** Relative intensity of the reaction intermediates normalized by CO$_2$ signal based on the results of (**a**) and (**b**). **d** MS signals for the pulse experiments of methyl formate and methyl formate-water (1:2) over 4.25Cu/Cu(Al)O$_x$ from 150 to 240 °C, respectively. **e** Schematic illustration for MSR reaction mechanism (HCOOCH$_3$* route).

(2800−3000 cm$^{-1}$) and very weak C=O bond attributed to HCHO or HCOOCH$_3$ species (1741 and 1770 cm$^{-1}$) are observed[7,8,24,26,36,38]. With the increase of ventilation time, the peak strength of these species enhances gradually, indicating that CH$_3$OH molecule undergoes activation and dehydrogenation on the catalyst surface. Subsequently, switching CH$_3$OH/He to H$_2$O/He (blue lines in Fig. 5a, b) results in the decline of these bands, demonstrating the transformation of these reaction intermediates after the introduction of H$_2$O. Furthermore, when exposing 4.25Cu/Cu(Al)O$_x$ catalyst to CH$_3$OH/H$_2$O/He from 50 to 270 °C (Fig. 5e), the signals of $\delta_{C-H}$, $\nu_{HCOO}$ and $\nu_{C=O}$ increase gradually, accompanied with the appearance of gas CO$_2$ (2380−2307 cm$^{-1}$)[12,26,32] and OH$^-$ group (3390 and 3723 cm$^{-1}$) (Supplementary Fig. 40), in accordance with the pulse experiments results (Fig. 4b).

Subsequently, we carried out in situ FT-IR to study the adsorption and reaction behavior of HCOOCH$_3$ intermediate at 240 °C on the

4.25Cu/Cu(Al)O$_x$ catalyst (Fig. 5c, green lines). After the introduction of HCOOCH$_3$, the bands attributed to C=O bond (1741 and 1768 cm$^{-1}$) in HCOOCH$_3$ and the ones assigned to C−H and COO$^-$ group from adsorbed HCOOCH$_3$ (1348, 1449 and 1582 cm$^{-1}$) are observed. After switching to a saturated water vapor (Fig. 5c, red lines), the C=O bond disappears accompanied with the gradual decline of HCOO* and CH$_3$O* species, corresponding to the HCOOCH$_3$ hydrolysis and C−H bond cleavage in reaction intermediates. In the case of in situ FT-IR measurement on the adsorption and reaction process of HCHO intermediate on 4.25Cu/Cu(Al)O$_x$ catalyst (Fig. 5d, gray lines), the intense absorption bands of HCOO* and CH$_3$O* (1459, 1349 and 1581 cm$^{-1}$) are detected, accompanied with the appearance of a weak C=O bond (1742 cm$^{-1}$), which is similar as the FT-IR spectra of adsorbed CH$_3$OH and HCOOCH$_3$ (Fig. 5a−c). Afterwards, a switching to saturated water vapor induces the decline of HCOO* and CH$_3$O* signals (Fig. 5d,

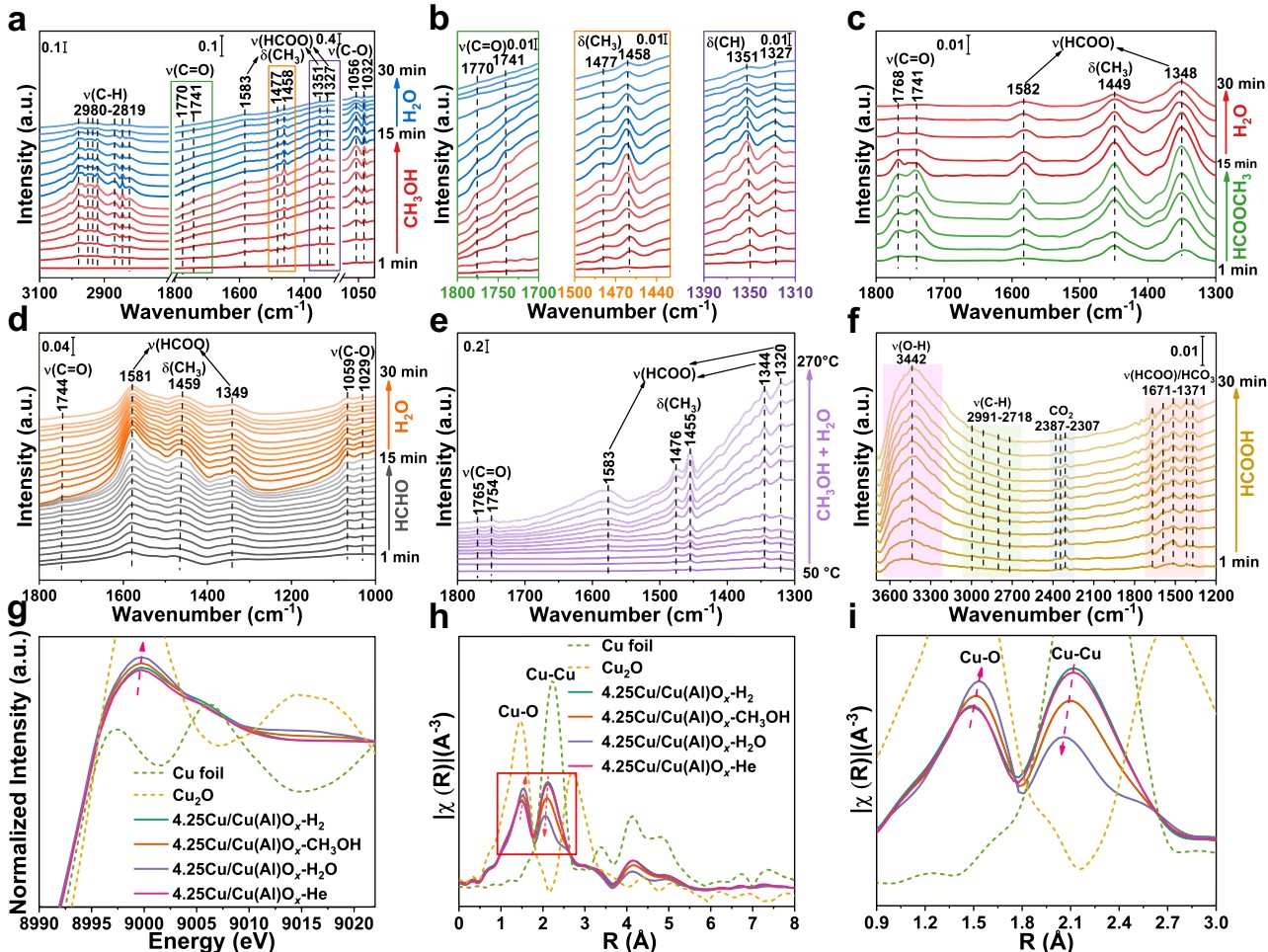

**Fig. 5 | In situ studies on active sites identification.** In situ FT-IR spectra of 4.25Cu/Cu(Al)O$_x$ along with the sequential introduction of **a**, **b** CH$_3$OH/He (1−15 min) and H$_2$O/He (15−30 min), **c** HCOOCH$_3$/He (1−15 min) and H$_2$O/He (15−30 min), **d** HCHO/He (1−15 min) and H$_2$O/He (15−30 min) at 240 °C, **e** CH$_3$OH/ H$_2$O/He from 50 to 270 °C, **f** HCOOH/He (1−30 min) at 180 °C. *Operando* XAFS spectra of Cu k-edge at **g** E-space and **h**, **i** R-space for pristine 4.25Cu/Cu(Al)O$_x$ catalyst (reduction in H$_2$, CH$_3$OH pumping, H$_2$O pumping and He purging in turn).

orange lines). The results above verify that the HCOOCH$_3$ intermediate is derived from the further transformation of formaldehyde, and water promotes the hydrolysis and transformation of methyl formate, which is consistent with the pulse experiment results (Fig. 4d).

To elucidate the Cu$^0$−Cu$^+$ synergistic catalytic mechanism, the CH$_3$OH conversion over 0.95Cu/Cu(Al)O$_x$ and 7.18Cu/Cu(Al)O$_x$ samples with the highest proportion of Cu$^+$ and Cu$^0$ respectively, was detected through in situ FT-IR (Supplementary Figs. 41, 42) combined with pulse experimental analysis (Supplementary Fig. 39). In comparison with the 4.25Cu/Cu(Al)O$_x$ catalyst, much stronger signals of CH$_3$O* and HCOO* are observed in the presence of 0.95Cu/Cu(Al)O$_x$ or 7.18Cu/Cu(Al)O$_x$, indicating a slower conversion rate. Moreover, relative to 4.25Cu/Cu(Al)O$_x$ (Fig. 5b), the IR absorption frequencies of CH$_3$O* and HCOO* intermediates show a blue-shift (move towards high wavenumber) and red-shift (move towards low wavenumber) over 0.95Cu/Cu(Al)O$_x$ (Supplementary Fig. 41b) and 7.18Cu/Cu(Al)O$_x$ (Supplementary Fig. 42b), respectively. In addition, in situ FT-IR measurements for CD$_3$OD conversion were also performed to study the isotope effects of C−D bonds cleavage in CD$_3$O* and DCOO* intermediates over the three catalysts (Supplementary Figs. 43−45). After switching to a saturated H$_2$O vapor, the consumption rates of CH$_3$O* ($R_{H_m}$) and HCOO* species ($R_{H_f}$) relative to CD$_3$O* ($R_{D_m}$) and DCOO* species ($R_{D_f}$) were calculated, where the $R_{H_m}$, $R_{H_f}$, $R_{D_m}$ and $R_{D_f}$ were the absolute value of slope from linear fitting (Supplementary Fig. 46 and Supplementary Note 3). The slight isotopic effect indicates that the

C−H bonds breakage is significantly promoted on the surface of 4.25Cu/Cu(Al)O$_x$ catalyst. The favorable transformation of HCOO* over 4.25Cu/Cu(Al)O$_x$ (Fig. 5f) is also demonstrated based on in situ FT-IR spectra of HCOOH adsorption: the C−H stretching vibration bands within 2700−3000 cm$^{-1}$ are inconspicuous in comparison with the 0.95Cu/Cu(Al)O$_x$ (Supplementary Fig. 47) and 7.18Cu/Cu(Al)O$_x$ samples (Supplementary Fig. 48). In addition, H$_2$/D$_2$ exchange (Supplementary Fig. 49) and H$_2$-TPD measurements (Supplementary Fig. 50 and Supplementary Note 4) demonstrate that the detachment of H from the catalyst surface is facile, where the Cu$^0$ site with a strong dehydrogenation capacity is responsible for the extraction of atomic H and desorption of H$_2$.

*Operando* XAFS measurements were carried out to reveal the dynamic evolution in electronic structure and coordination state of Cu$^0$−Cu$^+$ synergistic sites during the MSR reaction. As shown in Fig. 5g, compared with the pristine 4.25Cu/Cu(Al)O$_x$ sample (green line), the intensity of the white line peak increases along with the sequential introduction of CH$_3$OH (orange line) and H$_2$O (blue line), indicating that the Cu species undergoes an electronic reconfiguration due to the electron transfer from catalyst to reactive species. Correspondingly, both the coordination number and bond length of Cu−Cu decrease after the introduction of CH$_3$OH and H$_2$O, as demonstrated by the EXAFS spectra (Fig. 5h, i) and fitting results in R-space of Cu k-edge (Supplementary Figs. 51, 52 and Supplementary Table 5). In contrast, the coordination number and bond length of Cu−O increase

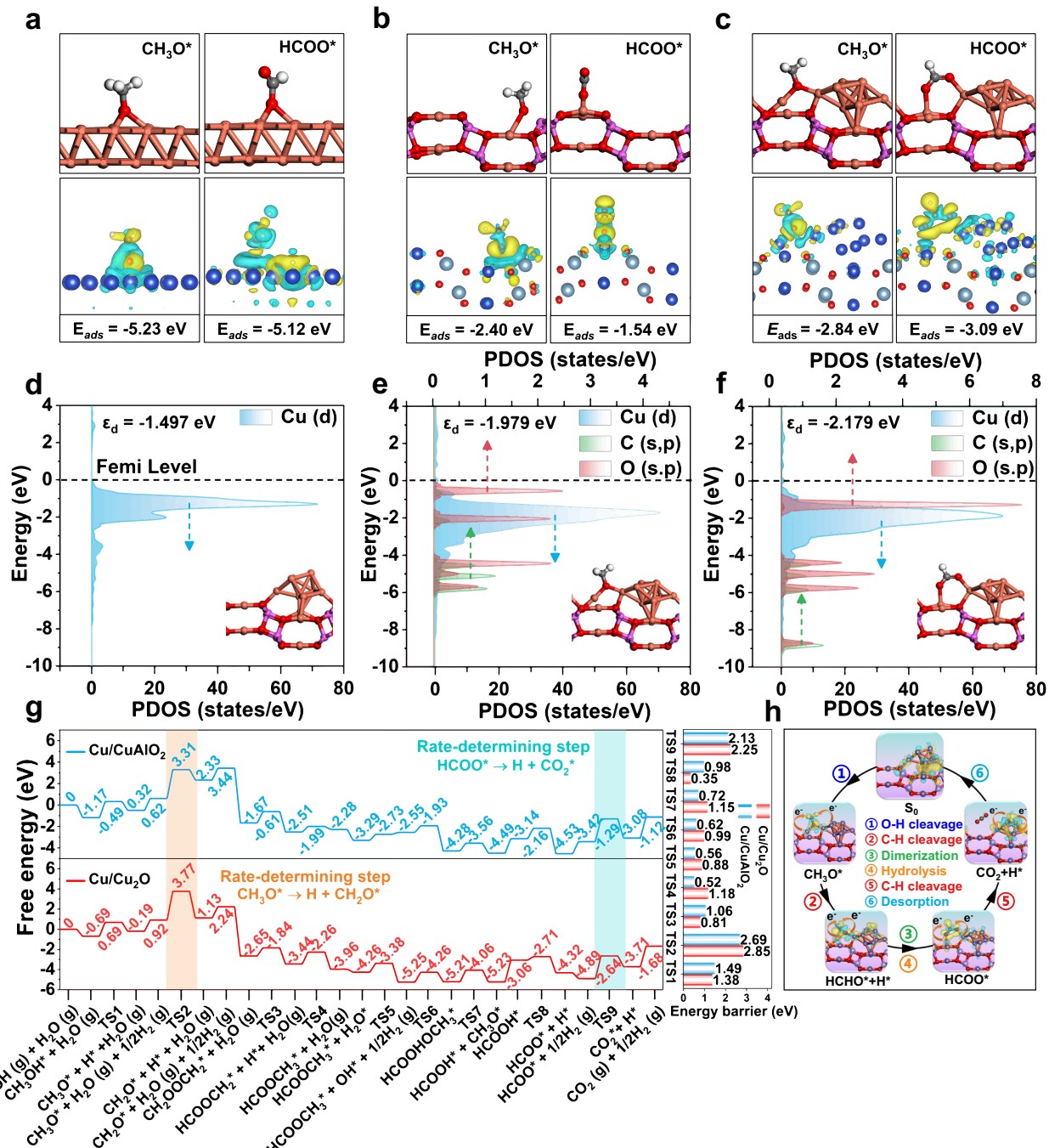

**Fig. 6 | DFT calculations on catalytic active sites towards MSR. a–c** Optimized adsorption configurations of CH₃O* and HCOO* accompanied with the charge density difference (CDD) on the **a** Cu(111), **b** CuAlO₂(101) and **c** Cu(111)/CuAlO₂(101) models, respectively (white, gray, red, orange and purple balls indicate H, C, O, Cu and Al atoms, respectively). **d–f** Projected density of states (PDOS) for the pristine **d** Cu/CuAlO₂, and adsorbed **e** CH₃O* and **f** HCOO* intermediates at the Cu/CuAlO₂ interface. **g** Full potential reaction pathway of MSR reaction following the HCOOCH₃* mechanism over Cu/CuAlO₂ and Cu/Cu₂O, respectively. 'TS' denotes the transition state. Numbers located at the horizontal line represent free energy of corresponding intermediate (the inset gives the energy barrier of each transient state). **h** Schematic illustration for the dynamic reconstruction process of Cu⁰–Cu⁺ interfacial sites towards C–H bonds cleavage of CH₃O* and HCOO*.

simultaneously, owing to the formation of additional Cu⁰–H and Cu⁺–O bonds. After He purging (pink line) for 15 min to remove the surface adsorbates, both the XANES and EXAFS spectra of 4.25Cu/Cu(Al)Oₓ restore to their initial states. The results substantiate that the Cu⁰–Cu⁺ interfacial sites participate in the substrate activation and C–H bonds cleavage. When *operando* XAFS measurements is performed by introducing HCHO (Supplementary Fig. 53), HCOOCH₃ (Supplementary Fig. 54) or HCOOH (Supplementary Fig. 55), a similar change is also observed, verifying a dynamic reconstruction process of the

Cu⁰–Cu⁺ interfacial sites during the whole MSR reaction. As a comparison, the 7.18Cu/Cu(Al)Oₓ sample does not show obvious variation in electronic and geometric structure (Supplementary Figs. 56, 57 and Supplementary Table 6), since a rather low ratio of Cu⁺/Cu⁰ is not conducive to C–H bonds cleavage.

Furthermore, the full-path potential energy barriers for MSR were studied by DFT calculations (Fig. 6 and supplementary Figs. 58–100). Given the results in XRD patterns and the stabilizing effect of amorphous alumina on Cu⁺, the optimized Cu(111)/Cu₂O(111) and Cu(111)/

$CuAlO_2(101)$ structures after thermodynamic analysis are used to model the $Cu^0$–$Cu^+$ interfacial sites (Supplementary Figs. 59 and 60). The calculated projected density of states (PDOS) (Supplementary Fig. 61 and Supplementary Note 5) confirm that the $Cu(111)/CuAlO_2(101)$ structure induces a remarkable electron coupling, whose $d$-band center ($\varepsilon_d = -1.497$ eV) is located between $Cu(111)$ ($\varepsilon_d = -0.212$ eV) and $CuAlO_2(101)$ ($\varepsilon_d = -1.618$ eV). A similar result is obtained in the $Cu(111)/Cu_2O(111)$ model, corresponding to the formation of $Cu^0$–$Cu^+$ interfacial sites, where the $d$-band center ($\varepsilon_d = -1.622$ eV) of $Cu(111)/Cu_2O(111)$ is located between $Cu(111)$ ($\varepsilon_d = -0.212$ eV) and $Cu_2O(111)$ ($\varepsilon_d = -1.722$ eV).

The Gibbs free energy diagrams and detailed schematic representation of the stepwise reaction for $HCOOCH_3*$ route on $Cu/Cu_2O$ and $Cu/CuAlO_2$ are displayed in Fig. 6g. Distinctly, the activation barriers for $CH_3O*$ formation (1.49 eV on $Cu/CuAlO_2$ and 1.38 eV on $Cu/Cu_2O$: $CH_3OH* \rightarrow CH_3O* + H*$) (Supplementary Figs. 62 and 77), $H_2O$ dissociation (0.56 eV on $Cu/CuAlO_2$ and 0.88 eV on $Cu/Cu_2O$: $H_2O* \rightarrow OH* + H*$) (Supplementary Figs. 64, 68, 79, and 83), $HCOOCH_3*$ formation (1.06 eV on $Cu/CuAlO_2$ and 1.18 eV on $Cu/Cu_2O$: $2CH_2O* \rightarrow HCOOCH_3*$) (Supplementary Figs. 66, 67, 81, 82 and Supplementary Note 6) and $HCOOCH_3*$ hydrolysis (0.72 eV on $Cu/CuAlO_2$ and 1.15 eV on $Cu/Cu_2O$: $HCOOCH_3* + OH* \rightarrow HCOOH* + CH_3O*$) (Supplementary Figs. 69, 70, 84, and 85) are lower than the dehydrogenation reactions of $CH_3O*$ (2.69 eV on $Cu/CuAlO_2$ and 2.85 eV on $Cu/Cu_2O$: $CH_3O* \rightarrow CH_2O*$) (Supplementary Figs. 63 and 78) and $HCOO*$ (2.13 eV on $Cu/CuAlO_2$ and 2.25 eV on $Cu/Cu_2O$: $HCOO* \rightarrow CO_2* + H*$) (Supplementary Figs. 72 and 87), indicating that the cleavage of C–H bonds in $CH_3O*$ and $HCOO*$ acts as the rate-determining step. This is consistent with the experimental results. In addition, we also studied the formaldehyde oxidation route (Supplementary Figs. 73–76) and $HCOOCH_3*$ route (Fig. 6g and Supplementary Figs. 62–72) on the $Cu/CuAlO_2$ model to reveal the role of $H_2O$ in the MSR reaction. Notably, for the $HCOOCH_3*$ pathway, $H_2O$ participates in the hydrolysis of methyl formate to form $CH_3O*$ and $HCOO*$ (Supplementary Figs. 68–70) with a reactively low energy barrier (0.72 eV). In contrast, for the formaldehyde oxidation pathway (Supplementary Figs. 73–75), the hydroxyl species from $H_2O$ dissociation oxidizes $CH_2O$ to $CH_2OOH$, followed by $CH_2OOH$ dehydrogenation to generate $HCOO$ with a high energy barrier of 4.20 eV. Thus, the methyl formate pathway is favorable on surface of $Cu/CuAlO_2$, and the role of water molecule is to hydrolyze methyl formate to produce formate, which is in accordance with the experimental results.

To in-depth study the $Cu^0$–$Cu^+$ synergistic catalysis mechanism, we further compared the reaction characteristics of rate-determining step ($CH_3O* \rightarrow CH_2O*$ and $HCOO* \rightarrow CO_2$) in $Cu^0$ ($Cu(111)$), $Cu^+$ ($Cu_2O$ and $CuAlO_2$) and $Cu^0$–$Cu^+$ model ($Cu/Cu_2O$ and $Cu/CuAlO_2$), respectively. The optimal adsorption configuration and charge density difference (CDD) (Fig. 6a–c, Supplementary Figs. 88, 89 and Supplementary Note 7) of $CH_3O*$ and $HCOO*$ are calculated, whose adsorption energies give the following order: $Cu(111)$ (−5.23 and −5.12 eV) > $Cu/Cu_2O$ (−3.37 and −4.47 eV) > $Cu/CuAlO_2$ (−2.84 and −3.09 eV) > $Cu_2O$ (−2.41 and −2.39 eV) > $CuAlO_2$ (−2.40 and −1.54 eV). In the case of $Cu(111)/CuAlO_2(101)$ system (Fig. 6c), the oxygen atom in $CH_3O*$ is co-adsorbed at the $Cu^0$–$Cu^+$ interfacial sites; whilst for $HCOO*$, the two oxygen atoms are adsorbed at interfacial $Cu^0$ and $Cu^+$ sites, respectively. These unique oxygen-end bridge adsorption configurations at the $Cu^0$–$Cu^+$ interfacial sites of $Cu/CuAlO_2$ confer a moderate adsorption strength of both intermediates, in agreement with the in situ FT-IR results.

Compared with the $Cu(111)$ ($Cu^0$ site), $CuAlO_2(101)$ ($Cu^+$ site) and $Cu_2O(111)$ ($Cu^+$ site), the dehydrogenation processes ($CH_3O* \rightarrow CH_2O*$ and $HCOO* \rightarrow CO_2$) involved in the rate-determining step are greatly boosted at the $Cu^0$–$Cu^+$ interfacial sites in the cases of $Cu(111)/Cu_2O(111)$ and $Cu(111)/CuAlO_2(101)$, especially for the latter system (Supplementary Fig. 96). As shown in Supplementary Figs. 63, 78, 90, 92, and 94, the energy barrier of $CH_3O*$ dehydrogenation follows the sequence: $CuAlO_2$ (4.07 eV) > $Cu$ (3.59 eV) > $Cu_2O$ (3.06 eV) > $Cu/Cu_2O$ (2.85 eV) > $Cu/CuAlO_2$ (2.69 eV). As shown in Supplementary Figs. 72, 87, 91, 93, and 95, the energy barrier of $HCOO*$ dehydrogenation gives the following order: $CuAlO_2$ (3.61 eV) > $Cu$ (2.92 eV) > $Cu_2O$ (2.74 eV) > $Cu/Cu_2O$ (2.25 eV) > $Cu/CuAlO_2$ (2.13 eV). The DFT calculation results support the experimental observations, in which the $Cu^0$–$Cu^+$ interfacial sites as intrinsic active centers facilitate the activation of reaction intermediates and promote the extraction of hydrogen, accounting for the extraordinarily high catalytic activity of $4.25Cu/Cu(Al)O_x$.

Furthermore, in terms of electronic structure, the adsorption of reaction intermediates ($CH_3O*$ and $HCOO*$) on the surface of $Cu/CuAlO_2$ induces the decrease of $d$-band center of $Cu$, accompanied with the occupied orbital energy moving away from the Fermi level (Fig. 6d–f), which indicates electron transfer from $d$-states of $Cu$ species to $CH_3O*$ and $HCOO*$. According to the Bader charge analysis (Supplementary Figs. 97 and 98), the $Cu^0$–$Cu^+$ interfacial sites in $Cu(111)/CuAlO_2(101)$ with oxyphilic ability result in the charge transfer from the catalyst interface to the adsorbed reaction intermediates. In terms of geometric structure, the bond length of Cu–O and Cu–Cu at the $Cu(111)/CuAlO_2(101)$ interface tends to elongate and shorten respectively during the process of C–H bonds activation (Supplementary Figs. 99 and 100), in good accordance with the reconfiguration phenomena obtained from in situ XAFS spectra. Thus, the dynamic evolution of electronic and geometric structure of $Cu^0$–$Cu^+$ interfacial sites towards C–H bonds cleavage is clearly revealed (Fig. 6h).

## Discussion

In summary, we report a $yCu/Cu(Al)O_x$ catalyst with well-defined and tunable $Cu^0$–$Cu^+$ interfacial sites applied to MSR reaction. The optimal catalytic performance is obtained on the $4.25Cu/Cu(Al)O_x$ catalyst with an appropriate $Cu^0$–$Cu^+$ interfacial sites, with a methanol conversion above 99%, a $H_2$ production rate of 110.8 μmol s$^{-1}$ g$_{cat}^{-1}$ and a satisfactory stability at 240 °C within 300 h. The MSR reaction over $4.25Cu/Cu(Al)O_x$ catalyst follows the $HCOOCH_3*$ route, and the $Cu^0$–$Cu^+$ interfacial synergistic catalysis plays a decisive role. The oxygen-containing intermediates ($CH_3O*$ and $HCOO*$) undergo activation adsorption at the $Cu^0$–$Cu^+$ interfacial sites with a moderate strength, giving rise to a reconstruction of catalyst interface as well as electron transfer from catalyst interface to reaction intermediates. The variations in both geometric and electronic structure result in a decreased energy barrier of C–H bonds fracture (the rate-determining step). This work provides atomic-level insights into $Cu^0$–$Cu^+$ interfacial synergistic catalysis in MSR, which can be extended to other heterogeneous catalytic systems towards rational design of high-performance catalysts.

## Methods

### Chemicals and materials
$Cu(NO_3)_2 \cdot 3H_2O$, $Al(NO_3)_3 \cdot 9H_2O$, $\gamma$-$Al_2O_3$, NaOH, $Na_2CO_3$ and $CH_3OH$ were obtained from the Aladdin chemical reagent company. Methanol-D4 ($CD_3OD$), Deuteromethanol-D ($CH_3OD$) and Deuterium oxide ($D_2O$) were purchased from Adamas-Beta chemical reagent company. Quartz sand ($SiO_2$, 40–60 mesh) was purchased from Tianjin Guangfu Fine Chemical Research Institute. Deionized (DI) water (resistivity: 18.2 MΩ·cm) was used in all experimental processes. All reagents were analytical grade and used without further purification.

### Synthesis of catalysts
The $yCu/Cu(Al)O_x$ catalysts were prepared via a co-precipitation method followed by the subsequent roasting and reduction processes. Typically, $Cu(NO_3)_2 \cdot 3H_2O$ (4.832 g) and $Al(NO_3)_3 \cdot 9H_2O$ (1.876 g) were dissolved in 50 mL of DI water (Solution A); NaOH (3.200 g) with 1.6 times equivalent concentration of metal ion (total of $Cu^{2+}$ and $Al^{3+}$)

and $Na_2CO_3$ (2.120 g) with 2.0 times equivalent concentration of tri-valent metal ion ($Al^{3+}$) were dissolved in 100 mL of DI water (Solution B). With vigorous stirring, Solution A and B were dropwise added into a beaker with water (30 mL) maintaining a stable pH (9.3–9.4). The obtained slurry was transferred to an oil bath and aged at 95 °C for 8 h. The resulting precipitate was filtered, washed thoroughly and dried at 80 °C for 12 h, followed by a calcination at 500 °C in air for 4 h, to obtain the $4.25CuAlO_x$ catalyst. Other $yCuAlO_x$ precursor samples ($y$ = 0.95, 2.32, 3.06, 4.25, 5.27, 7.18, respectively, representing total molar ratio of Cu/Al from ICP-AES; $x$ denotes the amount of oxygen coordinated with Cu and Al) were synthesized via the similar method described above, except changing the feeding amount of $Cu(NO_3)_2\cdot3H_2O$, $Al(NO_3)_3\cdot9H_2O$, NaOH and $Na_2CO_3$. Prior to use, the $yCuAlO_x$ samples were activated in a mixture gas (25% $H_2/N_2$, 50 mL min$^{-1}$) at 220 °C for 2 h to obtain the final catalysts, which were denoted as $yCu/Cu(Al)O_x$ ($y$ = 0.95, 2.32, 3.06, 4.25, 5.27, 7.18). The $4.25Cu/Cu(Al)O_x$–250 and $4.25Cu/Cu(Al)O_x$–300 were obtained via reducing $4.25CuAlO_x$ precursor at 250 and 300 °C, respectively. The $4.25Cu/Cu(Al)O_x$–600, $4.25Cu/Cu(Al)O_x$–700 and $4.25Cu/Cu(Al)O_x$–800 catalysts were obtained through firstly roasting the $4.25CuAlO_x$ precursor at 600, 700 and 800 °C for 4 h, followed by activation in a mixture gas (25% $H_2/N_2$, 50 mL min$^{-1}$) at 220 °C for 2 h.

The control sample $4.20CuO/Al_2O_3$ was prepared through a wet impregnation method. $Cu(NO_3)_2\cdot3H_2O$ (7.55 g) was dispersed in an aqueous mixture containing $\gamma$-$Al_2O_3$ powder (4.00 g), followed by sonication for 1 h to obtain a homogeneous suspension. After aging at room temperature for 6 h followed by drying the solvent at 80 °C, the resulting precipitate was calcinated at 500 °C in air for 4 h to obtain the catalyst precursor. Finally, the precursor was activated in a mixture gas ($H_2/N_2$ = 1:3, 50 mL min$^{-1}$) at 220 °C for 2 h to obtain the $4.20Cu/Al_2O_3$ sample.

## Catalytic evaluations

Catalytic performance towards MSR was evaluated in a fix-bed reactor equipped with a stainless-steel tube (interior diameter: 10 mm). In a catalytic process, 250 mg of catalyst precursor ($yCuAlO_x$) mixed with 2.5 g of quartz sand was pretreated in 25% $H_2/N_2$ flow (50.0 mL min$^{-1}$) at 220 °C for 2 h. After the temperature was cooled to reaction temperature in $N_2$ atmosphere, a water and methanol mixture with molar ratio of 2 was fed into the reactor by an injection pump at a rate of 0.040 mL min$^{-1}$. The reactants mixing with He gas (50.0 mL min$^{-1}$) were evaporated at 140 °C before entering the reactor to avoid product condensation. The react temperature was monitored by K-type thermocouple. The products were analyzed online by Shimadzu GC-17A (TDX-01 and HP-PLOT/Q columns) equipped with both FID and TCD detectors. The methanol conversion ($X_{MeOH}$), $CO_2$ selectivity ($S_{CO_2}$) and $H_2$ production rate ($R_{H_2}$) were calculated as follows:

$$X_{MeOH} = \frac{F_{MeOH,in} - F_{MeOH,out}}{F_{MeOH,in}} \times 100\% \qquad (1)$$

$$S_{CO_2} = \frac{F_{CO_2}}{F_{MeOH,in}} \times 100\% \qquad (2)$$

$$R_{H_2} = \frac{F_{H_2}}{m} \qquad (3)$$

Where $F_{MeOH,in/out}$ is the molar flow rate of methanol at the inlet/outlet of the reactor, respectively; $F_{H_2}$ and $F_{CO_2}$ denote the molar flow rate of $H_2$ and $CO_2$ at the reactor outlet; $m$ is the catalyst mass.

## Reaction dynamics studies

For the measurement of activation energy ($E_a$), reaction order, reaction rate of $CH_3OH$ ($r_{CH_3OH}$), $CD_3OD$ (($r_{CD_3OD}$) and $CH_3OD$ ($r_{CH_3OD}$) as well as the kinetic isotope effect (KIE) over various catalysts were studied under kinetic control conditions (methanol conversion less than 20%). Typically, the catalyst (40–60 mesh, 0.01–0.10 g) and quartz sand ($SiO_2$, 40–60 mesh, equivalent volume of the catalyst) were mixed together and transferred into the reactor tube. Reaction conditions for activation energy ($E_a$) measurement were as follows: liquid feed of $CH_3OH/H_2O$ (S/C = 2) at 0.040–0.080 mL min$^{-1}$ with He carrier gas (50 mL min$^{-1}$) at 180–260 °C. The kinetic isotope effect (KIE) test conditions were the similar as those described above except the reaction temperature at 210 °C. For the determination of reaction order of $CH_3OH$ and $H_2O$, the initial partial pressure of $CH_3OH$ and $H_2O$ was tuned within 13–56 kPa and 7–28 kPa, respectively.

More detailed experimental characterizations and computational methods are described in the Supplementary Information.

## Data availability

The data that supports the results reported in this manuscript are provided in the Supplementary Information File and in the Source Data file. Additional data related to this study are available from the authors upon request. Source data are provided with this paper.

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

## Acknowledgements

This work was supported by the National Natural Science Foundation of China (22172006, 22102006, 22002007, 22272007, U19B6002 and 22288102), the National Key R&D Program of China (2021YFC2103500), the Beijing Natural Science Foundation (2212012), and the Fundamental Research Funds for the Central Universities (XK2022-12). The authors are thankful for the support of the BSRF (Beijing Synchrotron Radiation Facility) during the XAFS measurements at the beamline of 1W1B and 1W2B.

## Author contributions

H.M. performed the catalyst preparation and characterizations. H.M., Z.Y., and J.Z. performed the catalytic evaluations. H.M. and Y.Y. prepared the draft manuscript. T.S. and P.Y. performed the DFT theoretical calculations. L.Z., L.W., W.L., and Z.R. participated in the catalyst structure investigations. Y.Y., J.Z., F.-S.X., and M.W. designed the study, analyzed the data, and revised the manuscript.

## Competing interests

The authors declare no competing interests.
