## [Peer Review File · Nature Communications]

Designing Cu⁰–Cu⁺ dual sites for improved C–H bond fracture for methanol steam reformingReviewers' comments:

Reviewer #1 (Remarks to the Author):

A very interesting paper in a hot topic. There are some fundamental points that must be addressed by the authors.

1) Reading the information provided on pages 11 and 12, I don't see why the studied catalyst is special when compared to catalysts reported in the literature for methane wet reforming. Is it better in terms of activity or selectivity?

2) The image in Figure 2A points to big separate regions of copper and copper oxide. How representative is this image? Under reaction conditions the copper oxide could be reduced by methanol (Orozco et al, J Phys Chem C, 2021, 125, 1, 558–571). This point needs clarification.

3) Page 18, the presence of surface OH groups could modify the chemistry of CH₃O on the surface (Orozco et al, J Phys Chem C, 2021, 125, 1, 558–571). Depending on the temperature, water could facilitate the desorption of CH₃O as CH₃OH.

3) Page 23, after looking at the images in Figure 2, I wonder how big is the concentration of Cu(0)-Cu(I) interfacial sites in the catalyst surfaces.

Reviewer #2 (Remarks to the Author):

In this work, a detailed experimental and theoretical study of the role of Cu⁺-Cu₂O sites in the MSR reaction mechanism is presented combining DFT, spectroscopic and kinetic studies. According to these results, an adequate balance between both sites is necessary for an optima catalytic performance, which seems reasonable, in line with other studies in the literature (J. Environ. Chem. Eng., 10, 2022, 107676;). However, according to the authors, the reason why this work should be published in Nature Communications, is because it address a deep understanding of active sites, untangling the ambiguity usually found in the literature. It is true that the literature is quite controversial in the assignation of active sites, and in this direction the authors did a complete and extended work, however, important points remains unclear and need to be defined accurately. In addition, certain conceptual aspects need to be further evaluated, making this work, as it is, unsuitable for publication. In addition, the above publication need to be revised and contrasted by the authors.

The work of M. Wei and collaborators, has been focused on Cu/Cu(Al)O_x samples prepared by a co-precipitation method with variable Cu loading. Both XRD and TEM data reveal the formation of Cu nanoparticles, Cu₂O and amorphous alumina. However, in some parts of the work, the authors point to amorphous alumina stabilized Cu⁺ species, indicating a Cu-O -Al interaction, which introduce some confusion. Moreover, if we look at the DFT model used by the authors, it doesn't correspond to Cu₂O nanoparticles on an Al₂O₃ support. This rise important questions, specifically because it is known that

the Lewis acidity of Cu⁺ species is strongly influenced by their local environment. One possibility to clarify this point is IR of CO adsorption, which, in fact, has been done by the authors. However, IR-CO titration of Cu⁺ species at room temperature is not accurate at all. For a more detailed analysis of surface species, low temperature analysis is requested (Adv.Catal.,47, 2002, 307-511).

Another point, which need to be carefully revised by the authors, is the assignation of the IR-CO band at 2105-2107 cm⁻¹. This band has been attributed by the authors to Cu⁺ species, however in literature studies, IR frequencies at higher frequencies, around 2120 -2145 cm⁻¹, are reported for CO adsorbed at defective sites of Cu₂O particles, which predominate at small particle sizes, as the one reported in this work (Angew. Chem. Int. Ed., 58, 2019, 4276-4280; J. Phys. Chem. B: 2000, 104, 6001-6011; ACS Catal.,12, 2022, 3845-3857). Therefore, a precise description of the here reported Cu⁺ sites need to be included, which is relevant compared to literature studies and for their catalytic performance. Also the fact that some spectroscopic studies (for instance XRD, TEM,) have been done on samples exposed to air, (or transferred in glove box, like in XPS) need to be considered in the discussion, knowing the fast surface oxidation of Cu₀.

Other aspect, which deserves to be considered in detail, is the amount of Cu⁺ -Cu₀ species.

Decovolution of AES peak (Figure S38) is not correct. AES peaks are broad and cannot be used for quantitative analysis. Moreover AES is not a surface sensitive technique and cannot be used for evaluating surface processes. On the other hand, extinction coefficients need to be taken into consideration if IR data are used. Finally, the analysis of Cu⁺ species based on N₂O titration, and TPD-CO is not clear, and need to be discussed in detail.

Another important point, which need to be considered in more detail, is the role of water in the reaction mechanism. In this work, the role of water is not properly considered. On one hand, it is not included in the DFT study, while need to be considered. On the other hand, the authors contemplate only a two - step reaction path in mechanistic studies, being firstly the activation of methanol on the surface of the catalyst, then, the addition of water. They don't consider the addition of water in the methanol activation, in line with the literature, where water plays a key role in the oxidation of CH₃OH to HCOO* species. Thus, according to the results obtained by the authors, they conclude that the role of water is the transformation of intermediates. This need to be discussed more in detail and contrasted with literature studies.

In conclusion, the authors must interpret their data more precisely, before reaching any conclusion about active sites.

Beside these comments, there are other points, which need to be revised

Page 16, Blue and red shift of the IR frequencies over 0.95 Cu/Cu(Al)O_x and 7.18 Cu/Cu(Al)O_x samples. What it mean?

Page 20. Dynamic reconstruction process of the Cu⁺-Cu₀ interfacial sites during the whole MSR reaction. Please, what it mean.

Additional comments to the authors:

DRIFT and Raman are not quantitative techniques. Therefore, it is not appropriate at all, to discuss differences in peak intensities among different samples, unless an internal reference is used for spectra normalization.

High X-ray power in the XPS studies, (for instance the 300W used in this work) may result in photo-

reduction, specifically in case of copper species.

In XPS studies, a reliable way to determine the oxidation state of copper species is analysing the auger parameter.

Reviewer #3 (Remarks to the Author):

The authors describe a catalyst for methanol reforming where they claim a specific Cu-Cu(I) site that is responsible for the good performance of the catalyst. This is interesting work, with a significant amount of characterization work. Any yet I do not agree that their study meet their stated criteria of “ a detailed study via employing spatially and temporally-resolved operando characterization techniques, kinetic investigations as well as theoretical calculations is imperative to shed light on the intrinsic active sites...”. There seems to be a disconnect between the information learned from bulk characterization techniques (e.g. XRD, XAS) and the catalytically active sites on particles that are

I don't understand the comment: “Notably, compared with Cu₂O standard sample, the 2θ of Cu₂O(111) at 36.8° in these samples shifts to a higher diffraction angle, close to the CuAlO₂(101) lattice plane (PDF#40-1037), which indicates the formation of Cu⁺ stabilized by amorphous alumina” when the Rietveld refinement confirms the presence of bulk Cu₂O.

I don't understand the TPR data. For example, 0.95CuAlO_x shows only a small amount of reduction <220C. However, the Rietveld refinement of the sample 0.95Cu/Cu(Al)O_x shows 31% metallic Cu. How are these data reconciled? How was this Cu reduced?

It appears that the 220C reduction results in a kinetically controlled reduction of the different types of Cu present in the sample. How did the catalytic performance vary with a reduction temperature, say 50C below/above this temperature?

The size if the Cu and Cu₂O particles is very large – from 7 to 9 nm. How can there be a unique Cu/Cu(I) site between such large particles?

XAS methodology:

The authors state: “transmission mode with a standard Lytle ion chamber”. A Lytle ion chamber is used for fluorescence XAS measurements so the statement is incorrect.

The authors state: “For the oprando MSR reaction, based on the above pretreatment, a certain amount of CH₃OH was carefully evaporated into the in situ cell in He flow (30 mL min⁻¹) at 200 °C along with signal collection; afterwards, saturated water vapor was introduced into the in situ cell under similar conditions, for the collection of XAFS spectra.” This is insufficient detail to allow an interested researcher to duplicate their findings.

The authors state: “All the XAFS data were processed using Athena software package”. This is simply insufficient information as to how the XAS data were processed. What k-range was used for the FT, what R-range was used in the fitting, what was the value of S0₂ – and how was it determined.

The analysis of the XAS data is rudimentary at best, and must be more extensive for Nature Commun. Stating that the average valence state is between 0 and 1 conveys no useful information – that is known from the XPS and XRD, for example. At a minimum LCF analysis of the XANES data is needed.

The EXAFS modeling of the Cu₂O has 2 O at 1.81 Å and 12 Cu at 3 Å – yet these are not the values reported in Table S2

The authors state: “the longer bond length of Cu–O in γ Cu/Cu(Al)Ox samples (1.84 Å) relative to Cu₂O standard (1.81 Å) implicates a distorted tetrahedral structure due to the partial substitution of Cu by Al, which is possibly related to the formation of unique Cu–O–Al geometric coordination”. There is absolutely no justification for this in the data presented in the manuscript. Table S2 contains no error bars on the data so how can the authors claim that it is possible to claim that 1.81 and 1.84 Å are statistically different?

Moreover: “Notably, the coordination number of Cu–Cu bond in γ Cu/Cu(Al)Ox increases from 5.1 to 8.2 whilst that of Cu–O bond declines from 1.7 to 0.7 along with the increment of Cu/Al ratio, in accordance with the valence state distribution of Cu species in these samples.”. How are the CN’s from the XAS reconciled with the fractions of Cu and Cu₂O determined from the Rietveld refinement?

There is no such thing as quasi-in situ XPS.

Response to Reviewers

Reviewer #1

Comments:

A very interesting paper in a hot topic. There are some fundamental points that must be addressed by the authors.

(1) Reading the information provided on pages 11 and 12, I don't see why the studied catalyst is special when compared to catalysts reported in the literature for methane wet reforming. Is it better in terms of activity or selectivity?

Author reply: Thank you for this comment. For the methanol steam reforming reaction, how to achieve a high hydrogen production rate at a reaction temperature as low as possible is a great challenge. In this work, the 4.25Cu/Cu(Al)O_x catalyst with Cu⁰–Cu⁺ synergetic effect shows a methanol conversion of 99.5% and a H₂ production rate of 110.8 μmol s⁻¹ g_{cat}⁻¹ at 240 °C. This is the leading level among copper-based catalysts ever reported (Supplementary Table 3), and even exceeds most precious metal catalysts.

● **Page 11, Line 11: rephrase:** “the 4.25Cu/Cu(Al)O_x catalyst exhibits the highest CH₃OH conversion (99.5%) as well as H₂ production rate (110.8 μmol s⁻¹ g_{cat}⁻¹) at 240 °C, which is preponderant to the state-of-the-art copper-based catalysts for MSR (Supplementary Table 3) and even exceeds most precious metal catalysts.”

(2) The image in Figure 2a points to big separate regions of copper and copper oxide. How representative is this image? Under reaction conditions the copper oxide could be reduced by methanol (Orozco et al, J Phys Chem C, 2021, 125, 1, 558–571). This point needs clarification.

Author reply: Thank you for this comment. The marked region in Fig. 2a illustrates the interwoven interface of Cu and Cu₂O phases in the 4.25Cu/Cu(Al)O_x catalyst, which is further verified by the STEM and EELS analysis (Fig. 2d–l). The existence of Cu₂O phase is related to the stabilizing effect of alumina support on Cu⁺, which suppresses its further reduction. For other γCu/Cu(Al)O_x samples,

remarkable interwoven interface between Cu and Cu₂O is also found around Cu nanoparticles (Supplementary Fig. 20). We have improved the discussion in the revised manuscript.

According to the reference mentioned above, the introduction of methanol would result in the reduction of partially oxidized copper species. However, this phenomenon is not fully applicable for the MSR reaction: when H₂O and CH₃OH are introduced together, the oxidation state of Cu species on the catalyst surface shows a dynamic change process (*ACS Catal.* 2019, 9, 2922). For instance in this work, the valence distribution of Cu species reaches an equilibrium after a long-term reaction as demonstrated by XPS analysis on the fresh 4.25Cu/Cu(Al)O_x (Fig. 1c) and used 4.25Cu/Cu(Al)O_x catalyst (Supplementary Fig. 30b), where the ratios of Cu⁺/Cu⁰ are 0.59 and 0.55 in the former and latter case, respectively.

● **Page 10, Line 1: rephrase:** “As shown in STEM and EELS, the existence of Cu₂O phase (Fig. 2d–l) is related to the stabilizing effect of amorphous alumina support on Cu⁺, which suppresses its further reduction.”

(3) Page 18, the presence of surface OH groups could modify the chemistry of CH₃O on the surface (Orozco et al, J Phys Chem C, 2021, 125, 1, 558–571). Depending on the temperature, water could facilitate the desorption of CH₃O as CH₃OH.

Author reply: Thank you for this comment. In order to identify the effect of surface OH group on methanol conversion, we measured the reaction order of H₂O over the 4.25Cu/Cu(Al)O_x catalyst in the revised manuscript (Fig. 3g). Obviously, the reaction rate of methanol does not change significantly as the partial pressure of H₂O increases. Through data fitting, the reaction order of H₂O (0.013) corresponds to a zero-order reaction. In addition, the isotope kinetic tests (Fig. 3f) show that the KIE value of D₂O ranges in 1.09–1.36 for all these γ Cu/Cu(Al)O_x samples, indicating that the water activation is not involved in the rate-determining step of MSR reaction. DFT calculations further confirm this issue. Thus, the influence of surface hydroxyl group on methanol conversion can be excluded. The corresponding discussion has been added in the revised manuscript.

● **Page 12, Line 19: rephrase:** “In addition, we measured the reaction order of CH₃OH and H₂O over the 4.25Cu/Cu(Al)O_x catalyst (Fig. 3g). The reaction rate of methanol displays a positive relationship with the CH₃OH partial pressure, but does not change significantly along with the increase of H₂O

partial pressure. Through data fitting, the reaction orders of CH₃OH and H₂O are determined to be 0.485 and 0.013, respectively, which indicates that the cleavage of C–H bond in methanol is crucial whilst the H₂O activation is not involved in the rate-determining step of MSR reaction.”

- Fig. 3g has been supplemented in the revised manuscript.

Fig. 3g Reaction order of CH₃OH and H₂O over 4.25Cu/Cu(Al)O_x catalyst at 240 °C.

(4) Page 23, after looking at the images in Figure 2, I wonder how big is the concentration of Cu(0)-Cu(I) interfacial sites in the catalyst surfaces.

Author reply: This is a valuable suggestion to improve the manuscript. According to this comment, we have made a rough estimation of the interfacial perimeter length ($L_{\text{Cu}^0\text{-Cu}^+}$) according to the reported method (*ACS Catal.* 2022, 12, 1315; *Appl. Catal. B: Environ.* 2022, 325, 122329). The Cu particles were assumed to have a hemispherical shape hypothesis: $S = D_{\text{Cu}} \times N_{\text{A}} \times X_{\text{Cu}} / (M \times N_{\text{Cu}})$; $d = 6M / (\sigma \times \rho \times D_{\text{Cu}} \times N_{\text{A}})$; $L_{\text{Cu}^0\text{-Cu}^+} = 2S/d$, where X_{Cu} is the copper content measured by ICP-AES; M is the Cu atom weight (63.546 g mol⁻¹); D_{Cu} is the dispersion degree of total Cu; S is total surface area of Cu species; N_{Cu} is the number of surface Cu atoms in one square meter area (1.46×10^{19} m⁻²); σ is the area occupied by a surface Cu atom (6.85 Å² per atom); ρ denotes the density of metallic Cu (8.94 g cm⁻³), and N_{A} is Avogadro’s number (6.022×10^{23} mol⁻¹). The corresponding computational details have been added in the revised Supplementary Information.

According to the above calculations, the $L_{\text{Cu}^0\text{-Cu}^+}$ values of $\gamma\text{Cu}/\text{Cu}(\text{Al})\text{O}_x$ samples are 3.43, 7.52, 9.12, 10.39, 8.64 and 6.39 ($\times 10^9 \text{ m g}_{\text{cat}}^{-1}$), respectively. Remarkably, a linear correlation between methanol reaction rate and $L_{\text{Cu}^0\text{-Cu}^+}$ is displayed in Fig. 3i, which further demonstrates that the $\text{Cu}^0\text{-Cu}^+$ synergistic catalysis plays a dominant role in C–H bond cleavage. Corresponding discussions have been added in the revised manuscript.

● **Supplementary Information, Page 4, Line 9: rephrase:** “The interfacial perimeter length ($L_{\text{Cu}^0\text{-Cu}^+}$) was estimated based on the follow equations, in which the Cu particles were assumed to have a hemispherical shape:

$$S = D_{\text{Cu}} \times N_{\text{A}} \times X_{\text{Cu}} / (M \times N_{\text{Cu}}) \quad (1)$$

$$d = 6M / (\sigma \times \rho \times D_{\text{Cu}} \times N_{\text{A}}) \quad (2)$$

$$L_{\text{Cu}^0\text{-Cu}^+} = 2S/d \quad (3)$$

where X_{Cu} is the copper content measured by ICP-AES; M is the Cu atom weight (63.546 g mol⁻¹); D_{Cu} is the dispersion degree of total Cu; S is total surface area of Cu species; N_{Cu} is the number of surface Cu atoms in one square meter area ($1.46 \times 10^{19} \text{ m}^{-2}$); σ is the area occupied by a surface Cu atom (6.85 Å² per atom); ρ denotes the density of metallic Cu (8.94 g cm⁻³), and N_{A} is Avogadro’s number ($6.022 \times 10^{23} \text{ mol}^{-1}$).”

● **Page 13, Line 12: rephrase:** “Furthermore, a linear correlation between methanol reaction rate and $L_{\text{Cu}^0\text{-Cu}^+}$ is obtained (Fig. 3i). It is thus concluded that catalytic activity depends on the synergistic catalysis of Cu^0 and Cu^+ rather than a single active site, and the $\text{Cu}^0\text{-Cu}^+$ interfacial sites are imperative for boosting the rate-determining step in MSR reaction (C–H bond cleavage).”

● Fig. 3i has been supplemented in the revised manuscript.

Fig. 3i Linear fitting results of methanol reaction rate as a function of both Cu^0 (C_{Cu^0}) and Cu^+ (C_{Cu^+}) as well as Cu^0 - Cu^+ interfacial perimeter.

Reviewer #2

Comments:

In this work, a detailed experimental and theoretical study of the role of Cu^+ - Cu^0 sites in the MSR reaction mechanism is presented combining DFT, spectroscopic and kinetic studies. According to these results, an adequate balance between both sites is necessary for an optimal catalytic performance, which seems reasonable, in line with other studies in the literature (*J. Environ. Chem. Eng.*, 10, 2022, 107676). However, according to the authors, the reason why this work should be published in Nature Communications, is because it addresses a deep understanding of active sites, untangling the ambiguity usually found in the literature. It is true that the literature is quite controversial in the assignment of active sites, and in this direction the authors did a complete and extended work, however, important points remain unclear and need to be defined accurately. In addition, certain conceptual aspects need to be further evaluated, making this work, as it is, unsuitable for publication. In addition, the above publication needs to be revised and contrasted by the authors.

(1) The work of M. Wei and collaborators, has been focused on $\text{Cu}/\text{Cu}(\text{Al})\text{O}_x$ samples prepared by a co-precipitation method with variable Cu loading. Both XRD and TEM data reveal the formation of Cu nanoparticles, Cu_2O and amorphous alumina. However, in some parts of the

work, the authors point to amorphous alumina stabilized Cu⁺ species, indicating a Cu-O-Al interaction, which introduce some confusion. Moreover, if we look at the DFT model used by the authors, it doesn't correspond to Cu₂O nanoparticles on an Al₂O₃ support.

Author reply: Thank you very much for this comment. We apologize for the misunderstanding caused by the unclear description. In this work, we found that Cu, Cu₂O and amorphous alumina phase co-exist in these γ Cu/Cu(Al)O_x samples, and the ratio of Cu⁺/Cu⁰ increases with the increment of Al content (Fig. 1a). Compared with the standard Cu₂O(111) lattice plane (36.4°), the 2θ of Cu₂O(111) at 36.8° in these samples shifts to a higher diffraction angle and is located between Cu₂O(111) and CuAlO₂(101) (Fig. 1b), indicating that partial Cu⁺ atoms can be stabilized by amorphous Al₂O₃ at the interfacial sites to form a CuAlO₂-like structure. In contrast, for commercial γ -Al₂O₃ supported Cu sample (Cu/Al₂O₃), the Cu particle merely shows Cu⁰ characteristics after reduction, further indicating the significance of stabilizing effect of amorphous Al₂O₃ on Cu⁺. In addition, the longer bond length of Cu–O in γ Cu/Cu(Al)O_x (1.84 Å) samples relative to Cu₂O standard (1.81 Å) based on the EXAFS spectra (Fig. 1f) demonstrates a distorted tetrahedral structure owing to the interfacial Cu–O–Al coordination (CuAlO₂-like structure).

This result is further verified by DFT calculations (Fig. 1g–i), in which the Cu⁺–O bond length in Cu(111)/CuAlO₂(101) (1.90 Å) is significantly longer than that in Cu₂O(111) (1.83 Å) and Cu(111)/Cu₂O(111) (1.83 Å). Furthermore, according to the results of H₂-TPR (Supplementary Fig. 11), STEM and EELS (Fig. 2), we conclude that some Cu atoms at the Cu-Al₂O₃ interface are difficult to be completely reduced due to the stabilizing effect of amorphous alumina, which leads to the formation of Cu₂O phase at the edge of Cu nanoparticle. Similar phenomenon has also been observed in Cu/SiO₂ (*ACS Catal.* 2015, 5, 6200; *Chem. Sci.* 2019, 10, 2578; *Nat. Commun.* 2013, 4, 2339; *J. Am. Chem. Soc.* 2012, 134, 13922; *Angew. Chem. Int. Ed.* 2021, 60, 15344), Cu/CeO₂ (*ACS Catal.* 2022, 12, 1315; *Nat. Commun.* 2022, 13, 867), Cu/ZnO (*Appl. Catal. A: Gen.* 2021, 616, 118072), Cu/La₂O₃ (*Appl. Catal. B: Environ.* 2019, 251, 119), Cu/Y₂O₃ (*Res. Chem. Intermed.* 2022, 48, 3389) and Cu/Al₂O₃-SiO₂ (*Catal. Sci. Technol.* 2022, 12, 5879), where the strong interaction between oxide support and Cu particle stabilizes Cu⁺ species. Corresponding discussions have been supplemented in the revised manuscript.

In the DFT calculations, in order to study the stabilizing effect of amorphous Al₂O₃ on Cu₂O, an

ideal model was constructed, *i.e.*, CuAlO₂ (Al₂O₃ + Cu₂O) was used to simulate the Cu⁺ site. Taking into account the reviewer's suggestion, we also built the Cu(111)/Cu₂O(111) model and calculated the full potential reaction pathway of ESR in the revised manuscript (Fig. 6g and Supplementary Figs. 71–79). Corresponding discussions have been supplemented in the revised manuscript.

● **Page 5, Line 19: rephrase:** “Notably, compared with the standard Cu₂O(111) lattice plane (36.4°), the Cu₂O(111) reflection at 2θ 36.8° in these samples shifts to a higher diffraction angle (Fig. 1b) and is located between Cu₂O(111) and CuAlO₂(101) (PDF#40-1037), which indicates that partial Cu⁺ atoms can be stabilized by amorphous Al₂O₃ at the interfacial sites to form a CuAlO₂-like structure.”

● **Page 8, Line 11: rephrase:** “Based on the fitting results and wavelet transform (Supplementary Fig. 17 and Supplementary Table 2), the longer bond length of Cu–O in γ Cu/Cu(Al)O_x samples (1.84 Å) relative to Cu₂O standard (1.81 Å) indicates a distorted tetrahedral structure due to the partial substitution of Cu by Al, which is possibly related to the formation of unique Cu–O–Al geometric coordination (CuAlO₂-like structure). This result is consistent with the DFT calculation results (Fig. 1g–i), in which the Cu⁺–O bond length in Cu(111)/CuAlO₂(101) (1.90 Å) is significantly longer than that in Cu₂O(111) (1.83 Å) and Cu(111)/Cu₂O(111) (1.83 Å).”

● **Page 25, Line 4: rephrase:** “Compared with the Cu(111) (Cu⁰ site), CuAlO₂(101) (Cu⁺ site) and Cu₂O(111) (Cu⁺ site), the dehydrogenation processes (CH₃O* → CH₂O* and HCOO* → CO₂) involved in the rate-determining step are greatly boosted at the Cu⁰–Cu⁺ interfacial sites in the cases of Cu(111)/Cu₂O(111) and Cu(111)/CuAlO₂(101), especially for the latter system (Supplementary Fig. 88). As shown in Supplementary Figs. 59,72,82,84,86, the energy barrier of CH₃O* dehydrogenation follows the sequence: CuAlO₂ (4.07 eV) > Cu (3.59 eV) > Cu₂O (3.06 eV) > Cu/Cu₂O (2.85 eV) > Cu/CuAlO₂ (2.69 eV). As shown in Supplementary Figs. 66,79,83,85,87, the energy barrier of HCOO* dehydrogenation gives the following order: CuAlO₂ (3.61 eV) > Cu (2.92 eV) > Cu₂O (2.74 eV) > Cu/Cu₂O (2.25 eV) > Cu/CuAlO₂ (2.13 eV). The DFT calculation results support the experimental observations, in which the Cu⁰–Cu⁺ interfacial sites as intrinsic active centers facilitate the activation of reaction intermediates and promote the extraction of hydrogen, accounting for the extraordinarily high catalytic activity of 4.25Cu/Cu(Al)O_x.”

● Fig. 1g–i and Fig. 6g have been supplemented in the revised manuscript.

Fig. 1 Cu–O bond length in **g** Cu₂O(111), **h** Cu(111)/Cu₂O(111) and **i** Cu(111)/CuAlO₂(101) systems based on DFT calculations (red: O; orange: Cu; purple: Al).

Fig. 6g Full potential reaction pathway of ESR reaction following the HCOOCH₃* mechanism over Cu/CuAlO₂ and Cu/Cu₂O, respectively. ‘TS’ denotes the transition state. Numbers located at the horizontal line represent free energy of corresponding intermediate (the inset gives the energy barrier of each transient state).

• Supplementary Figs. 71–79 and 82–88 have been supplemented in the revised Supplementary Information.

Supplementary Figure 71. Calculated potential energy diagram and corresponding geometric configurations for the dehydrogenation of CH_3OH on the surface of $\text{Cu}(111)/\text{Cu}_2\text{O}(111)$ (*, IS, TS and FS represent the adsorption state, initial state, transition state and final state, respectively; E_a and ΔE is the energy barrier and thermodynamic energy).

Supplementary Figure 72. Calculated potential energy diagram and corresponding geometric configurations for the dehydrogenation of CH_3O^* on the surface of $\text{Cu}(111)/\text{Cu}_2\text{O}(111)$ (*, IS, TS and FS represent the adsorption state, initial state, transition state and final state, respectively; E_a and ΔE is the energy barrier and thermodynamic energy).

Supplementary Figure 73. Calculated potential energy diagram and corresponding geometric configurations for the dissociation of H_2O on the surface of $\text{Cu}(111)/\text{Cu}_2\text{O}(111)$ (*, IS, TS and FS represent the adsorption state, initial state, transition state and final state, respectively; E_a and ΔE is the energy barrier and thermodynamic energy).

Supplementary Figure 74. Calculated potential energy diagram and corresponding geometric configurations for the dehydrogenation of CH_2OOCH_3 on the surface of $\text{Cu}(111)/\text{Cu}_2\text{O}(111)$ (*, IS, TS and FS represent the adsorption state, initial state, transition state and final state, respectively; E_a and ΔE is the energy barrier and thermodynamic energy).

Supplementary Figure 75. Calculated potential energy diagram and corresponding geometric configurations for the dissociation of H_2O on the surface of $\text{Cu}(111)/\text{Cu}_2\text{O}(111)$ with the existence of HCOOCH_3 (*, IS, TS and FS represent the adsorption state, initial state, transition state and final state, respectively; E_a and ΔE is the energy barrier and thermodynamic energy).

Supplementary Figure 76. Calculated potential energy diagram and corresponding geometric configurations for the hydrolysis of HCOOCH_3 on the surface of $\text{Cu}(111)/\text{Cu}_2\text{O}(111)$ (*, IS, TS and FS represent the adsorption state, initial state, transition state and final state, respectively; E_a and ΔE is the energy barrier and thermodynamic energy).

Supplementary Figure 77. Calculated potential energy diagram and corresponding geometric configurations for the dissociation of HCOOHOCH_3 to HCOOH and CH_3O^* on the surface of $\text{Cu}(111)/\text{Cu}_2\text{O}(111)$ (*, IS, TS and FS represent the adsorption state, initial state, transition state and final state, respectively; E_a and ΔE is the energy barrier and thermodynamic energy).

Supplementary Figure 78. Calculated potential energy diagram and corresponding geometric configurations for the dehydrogenation of HCOOH on the surface of $\text{Cu}(111)/\text{Cu}_2\text{O}(111)$ (*, IS, TS and FS represent the adsorption state, initial state, transition state and final state, respectively; E_a and ΔE is the energy barrier and thermodynamic energy).

Supplementary Figure 79. Calculated potential energy diagram and corresponding geometric configurations for the dehydrogenation of HCOO^* on the surface of $\text{Cu}(111)/\text{Cu}_2\text{O}(111)$ (*, IS, TS and FS represent the adsorption state, initial state, transition state and final state, respectively; E_a and ΔE is the energy barrier and thermodynamic energy).

Supplementary Figure 82. Calculated potential energy diagram and corresponding geometric configurations for the dehydrogenation of CH_3O^* on the surface of $\text{Cu}(111)$ (*, IS, TS and FS represent the adsorption state, initial state, transition state and final state, respectively; E_a , and ΔE is the energy barrier and thermodynamic energy).

Supplementary Figure 83. Calculated potential energy diagram and corresponding geometric configurations for the dehydrogenation of HCOO^* on the surface of $\text{Cu}(111)$ (*, IS, TS and FS represent the adsorption state, initial state, transition state and final state, respectively; E_a , and ΔE is the energy barrier and thermodynamic energy).

Supplementary Figure 84. Calculated potential energy diagram and corresponding geometric configurations for the dehydrogenation of CH_3O^* on the surface of $\text{CuAlO}_2(101)$ (*, IS, TS and FS represent the adsorption state, initial state, transition state and final state, respectively; E_a , and ΔE is the energy barrier and thermodynamic energy).

Supplementary Figure 85. Calculated potential energy diagram and corresponding geometric configurations for the dehydrogenation of HCOO^* on the surface of $\text{CuAlO}_2(101)$ (*, IS, TS and FS represent the adsorption state, initial state, transition state and final state, respectively; E_a , and ΔE is the energy barrier and thermodynamic energy).

Supplementary Figure 86. Calculated potential energy diagram and corresponding geometric configurations for the dehydrogenation of CH_3O^* on the surface of $\text{Cu}_2\text{O}(111)$ (*, IS, TS and FS represent the adsorption state, initial state, transition state and final state, respectively; E_a , and ΔE is the energy barrier and thermodynamic energy).

Supplementary Figure 87. Calculated potential energy diagram and corresponding geometric configurations for the dehydrogenation of HCOO^* on the surface of $\text{Cu}_2\text{O}(111)$ (*, IS, TS and FS represent the adsorption state, initial state, transition state and final state, respectively; E_a , and ΔE is the energy barrier and thermodynamic energy).

Supplementary Figure 88. Reaction energy barrier of CH_3O^* and HCOO^* dehydrogenation over various models (the results from the Supplementary Figs. 59,66,72,79 and 82–87).

(2) This rise important questions, specifically because it is known that the Lewis acidity of Cu^+ species is strongly influenced by their local environment. One possibility to clarify this point is IR of CO adsorption, which, in fact, has been done by the authors. However, IR-CO titration of

Cu⁺ species at room temperature is not accurate at all. For a more detailed analysis of surface species, low temperature analysis is requested (*Adv. Catal.*, 47, 2002, 307-511). Another point, which need to be carefully revised by the authors, is the assignation of the IR-CO band at 2105-2107 cm⁻¹. This band has been attributed by the authors to Cu⁺ species, however in literature studies, IR frequencies at higher frequencies, around 2120 -2145 cm⁻¹, are reported for CO adsorbed at defective sites of Cu₂O particles, which predominate at small particle sizes, as the one reported in this work (*Angew. Chem. Int. Ed.*, 2019, 58, 4276-4280; *J. Phys. Chem. B.*, 2000, 104, 6001-6011; *ACS Catal.*,12, 2022, 3845-3857). Therefore, a precise description of the here reported Cu⁺ sites need to be included, which is relevant compared to literature studies and for their catalytic performance.

Author reply: Thank you for this comment. According to this comment, we carried out pyridine-IR and NH₃-TPD-MS to measure the Lewis acidity of these catalysts (Fig. R1). Remarkably, all these samples show similar surface acidity, and no significant correlation between methanol reaction rate and surface acid quantity can be found. In addition, although low temperature CO-IR provides fine CO absorption signals, room temperature CO-IR has already been employed to identify surface Cu⁺ and Cu⁰ species, and even used for semi-quantitative analysis (*Nat. Catal.* 2022, 5, 99; *Angew. Chem. Int. Ed.* 2021, 60, 15344; *J. Am. Chem. Soc.* 2021, 143, 2984; *ACS Catal.* 2020, 10, 14694; *Appl. Catal. B: Environ.* 2022, 313, 121468; *Catal. Today* 2017, 283, 134). According to the literature search and comparison, it is found that the CO–Cu⁺ IR signals on different catalysts show great differences, which are closely related to the metal element type and electronic structure of the catalyst. For instance, CO–Cu⁺ signal is located at 2127 cm⁻¹ for Cu/SiO₂ (*ACS Catal.* 2020, 10, 14694), 2140–2110 cm⁻¹ for Cu/Al₂O₃ (*Phys. Chem. B* 2000, 104, 25, 6001), 2092 cm⁻¹ for Cu/CeO₂ (*Nat. Commun.* 2022, 13, 867), 2081 cm⁻¹ for Cu/SiO₂ (*Angew. Chem. Int. Ed.* 2021, 60, 15344), 2108 cm⁻¹ for Cu/ZnO (*Nat. Commun.* 2021, 12, 4331), 2104–2106 cm⁻¹ for Cu/ZnAlO_x (*Nat. Catal.* 2022, 5, 99), 2108–2111 cm⁻¹ for Cu/CeO₂ (*Nat. Catal.* 2019, 2, 334), 2102 cm⁻¹ for Cu/ZnO (*J. Am. Chem. Soc.* 2021, 143, 2984), 2104 cm⁻¹ for Cu/CeO₂ (*J. Energy Inst.* 2022, 104, 142), 2113 cm⁻¹ for Cu/Al₂O₃ (*J. Catal.* 2014, 319, 127), 2098 cm⁻¹ for Cu/MgAlO_x (*J. Catal.* 2020, 385, 160), 2105 cm⁻¹ for Cu/Al₂O₃ (*Appl. Catal. B: Environ.* 2022, 313, 121468), 2104 cm⁻¹ for Cu/TiO₂ (*Catal. Commun.* 2016, 78, 33). Thus, the room temperature CO–Cu⁺ IR signal in this work is within a reasonable range compared with previous

reports. This is a good suggestion, and we will try to explore it in the future research.

Figure R1. a) pyridine-IR and b) NH₃-TPD-MS measurements for γ Cu/Cu(Al)O_x and control samples.

(3) Also the fact that some spectroscopic studies (for instance XRD, TEM) have been done on samples exposed to air, (or transferred in glove box, like in XPS) need to be considered in the discussion, knowing the fast surface oxidation of Cu⁰.

Author reply: Thank you for this comment. For the *off situ* characterizations, a passivation treatment of the catalyst samples was performed, so as to prevent the deep oxidation of sample in air. The relevant experimental details have been added in the revised Supplementary Information.

• **Supplementary Information, Page 3, Line 4: rephrase:** “For the *off situ* characterizations, after the reduction of catalyst precursor (γ CuAlO_x) in a tube furnace, the sample was passivated by 0.5% O₂/N₂ mixture at room temperature to produce a thin oxide layer, so as to prevent the deep oxidation of sample in air.”

(4) Other aspect, which deserves to be considered in detail, is the amount of Cu⁺-Cu⁰ species. Deconvolution of AES peak (Figure S38) is not correct. AES peaks are broad and cannot be used for quantitative analysis. Moreover, AES is not a surface sensitive technique and cannot be used for evaluating surface processes. On the other hand, extinction coefficients need to be taken into consideration if IR data are used. Finally, the analysis of Cu⁺ species based on N₂O titration, and TPD-CO is not clear, and need to be discussed in detail.

Author reply: Thank you for this comment. Although there are some deviations in AES spectra analysis, this technique is still an important quantitative analysis method for valence distribution of copper species on catalyst surface at present (*Chem. Sci.* 2019, 10, 2578; *ACS Catal.* 2022, 12, 1315; *ACS Catal.* 2017, 7, 7890; *ACS Catal.* 2015, 5, 6200; *Appl. Catal. A: Gen.* 2021, 616, 118072; *Nat. Commun.* 2013, 4, 2339).

According to this comment, for CO-IR measurements, we refer to the extinction coefficient reported in the literature (*Micropor. Mesopor. Mat.* 2012, 162, 175) for linear CO adsorption at Cu⁰ ($\epsilon_{\text{Cu}^0} = 0.79$) and Cu⁺ ($\epsilon_{\text{Cu}^+} = 1.30$) sites, and recalculated the Cu⁺/Cu⁰ ratio based on the following equation: $\text{Cu}^+/\text{Cu}^0 = (A_{\text{Cu}^+}/\epsilon_{\text{Cu}^+})/(A_{\text{Cu}^0}/\epsilon_{\text{Cu}^0})$. The results in Fig. 1d have been improved in the revised manuscript. In addition, the methods for quantitative analysis of surface concentrations of Cu⁰ and Cu⁺ by using the N₂O titration and CO-TPD were described in detail in the revised Supplementary Information.

• **Supplementary Information, Page 3, Line 7: rephrase:**

N₂O-titration measurements. The catalyst precursor ($\gamma\text{-CuAlO}_x$) was reduced in H₂ at 220 °C for 1 h, followed by purging with Ar for 0.5 h and cooling down to 50 °C. Then, the sample was exposed to 5% N₂O/Ar flow (50 mL min⁻¹) at 50 °C for 1 h for the oxidation of surface copper species to Cu₂O. Subsequently, the sample was ramped from 50 to 300 °C at a rate of 10 °C min⁻¹ in 25% H₂/Ar (50 mL min⁻¹). The consumed hydrogen amount (X) was calculated. The dispersion degree of Cu⁰ (D_{Cu^0}) and the concentration of surface Cu⁰ species (C_{Cu^0}) were calculated according to the following equations: $D_{\text{Cu}^0} = (2 \times M_{\text{Cu}} \times X / (m_{\text{cat}} \times m_{\text{Cu}})) \times 100\%$; $C_{\text{Cu}^0} = D_{\text{Cu}^0} \times m_{\text{Cu}} / M_{\text{Cu}}$ (m_{cat} is the catalyst mass; m_{Cu} is copper mass per unit mass of catalyst measured by ICP-AES; M_{Cu} is atomic mass of copper).

N₂O-CO TPD measurements. The catalyst precursor ($\gamma\text{-CuAlO}_x$) was firstly pretreated at 220 °C in a H₂ atmosphere for 1 h, followed by purging with Ar at 220 °C for 1 h. Then, the catalyst was exposed to 5% N₂O/Ar flow (50 mL min⁻¹) at 50 °C for 1 h for the oxidation of surface copper species to Cu₂O. After the temperature was decreased to 30 °C, 10 vol % CO with He as carrier gas was introduced until a saturation adsorption. Subsequently, pure He was purged at 30 °C to remove physically-adsorbed CO, and the temperature was increased from 30 to 500 °C at a rate of 10 °C min⁻¹ for the collection of signals. The amount of CO desorption value (A) corresponds to the amount of total

surface copper species. The dispersion degree of total surface copper ($\text{Cu}^+ + \text{Cu}^0$) species (D_{Cu}) and Cu^+ species (D_{Cu^+}) as well as the concentration of surface Cu^+ species (C_{Cu^+}) were calculated according to the following equations: $D_{\text{Cu}} = M_{\text{Cu}} \times A / (m_{\text{cat}} \times m_{\text{Cu}}) \times 100\%$; $D_{\text{Cu}^+} = D_{\text{Cu}} - D_{\text{Cu}^0}$; $C_{\text{Cu}^+} = D_{\text{Cu}^+} \times m_{\text{Cu}} / M_{\text{Cu}}$.

- Fig. 1d has been improved in the revised manuscript.

Fig. 1d CO-DRIFT spectra of $\gamma\text{Cu}/\text{Cu}(\text{Al})\text{O}_x$ and control samples.

(5) Another important point, which need to be considered in more detail, is the role of water in the reaction mechanism. In this work, the role of water is not properly considered. On one hand, it is not included in the DFT study, while need to be considered. On the other hand, the authors contemplate only a two-step reaction path in mechanistic studies, being firstly the activation of methanol on the surface of the catalyst, then, the addition of water. They don't consider the addition of water in the methanol activation, in line with the literature, where water plays a key role in the oxidation of CH_3OH to HCOO^* species. Thus, according to the results obtained by the authors, they conclude that the role of water is the transformation of intermediates. This need to be discussed more in detail and contrasted with literature studies. In conclusion, the authors must interpret their data more precisely, before reaching any conclusion about active sites.

Author reply: This is a valuable suggestion to improve the manuscript. To clarify the catalytic role of H_2O during the methanol reforming, we further determined reaction order of H_2O within dynamic

range. As shown in Fig. 3g, the methanol conversion does not show obvious change as the partial pressure of H₂O increases. Through data fitting, the reaction order of H₂O (0.013) corresponds to a zero-order reaction. In addition, the isotope kinetic tests (Fig. 3f) show that the KIE value of D₂O ranges in 1.09–1.36 for all these γ Cu/Cu(Al)O_x samples, indicating that the water activation is not involved in the rate-determining step of MSR reaction. DFT calculations further confirm this issue, where the energy barrier of H₂O dissociation (H₂O → OH⁻ + H⁺) (Supplementary Figs. 60,62,73,75) is much less than C–H bonds cleavage in CH₃O and HCOO over Cu/CuAlO₂ and Cu/Cu₂O.

In addition, we also carried out MS analysis for the *operando* pulse experiments when CH₃OH and CH₃OH + H₂O was introduced, respectively (Fig. 4a–c). In the absence of H₂O in reaction system, HCOOCH₃ is heavily generated accompanied with very little HCOO* signal. In contrast, when CH₃OH + H₂O is co-introduced, the relative intensity of HCOOCH₃ decreases accompanied by the increase of HCOO* signal. Moreover, in comparison with pulse experiments of HCOOCH₃ over 4.25Cu/Cu(Al)O_x catalyst (Fig. 4d), the co-introduction of H₂O (HCOOCH₃ + H₂O) greatly promotes the hydrolysis of HCOOCH₃, with significantly weakened HCOOCH₃ and enhanced CH₃O* and HCOO* signals.

Meanwhile, we monitored the signal change of *in situ* FT-IR with the introduction of CH₃OH + H₂O from 50 to 270 °C (Fig. 5e). The results show that the intensity of C=O in HCOOCH₃ and HCOO* in HCOOH increases with the increment of temperature. According to the kinetic studies (Fig. 3f, g), the fracture of C–H bonds in CH₃O* and HCOO* intermediates is proved as the rate-determining step, and water molecule promotes the decomposition of HCOOCH₃ intermediate but does not participate directly in the cleavage of C–H bonds. The corresponding discussions have been improved in the revised manuscript.

Furthermore, we carried out *in situ* FT-IR to study the adsorption and reaction behavior of HCOOCH₃ intermediate on the 4.25Cu/Cu(Al)O_x catalyst (Fig. 5c, green lines). After the introduction of HCOOCH₃, the bands attributed to C=O bond (1741 and 1768 cm⁻¹) in HCOOCH₃ and the ones assigned to C–H and COO⁻ group from adsorbed HCOOCH₃ (1348, 1449 and 1582 cm⁻¹) are observed. After switching to a saturated water vapor (Fig. 5c, red lines), the C=O bond disappears accompanied with the gradual decline of HCOO* and CH₃O* species, corresponding to the HCOOCH₃ hydrolysis and C–H bond cleavage in reaction intermediates. The corresponding discussions have been improved in the revised manuscript.

According to the reviewer's comments, we further discussed the role of water by DFT calculations in the revised manuscript, in which the two possible contributions of H₂O (HCOOCH₃ hydrolysis (Fig. 6g and Supplementary Figs. 61-64) and methoxy oxidation (Supplementary Figs. 67-70)) were studied. Notably, in the former case, H₂O mainly helps in the hydrolysis of methyl formate to form CH₃O* and HCOO* (Supplementary Figs. 62-64) with a reactively low energy barrier of 0.72 eV. In the latter case, the hydroxyl species from H₂O dissociation oxidizes CH₂O to HCOO with an extremely high energy barrier of 4.20 eV (Supplementary Figs. 67-69). Thus, we believe that the main role of water molecule is to hydrolyze methyl formate to produce formate, which follows the methyl formate mechanism. The corresponding discussions have been added in the revised manuscript.

● **Page 15, Line 12: rephrase:** “As shown in Fig. 4a, the signals of reaction intermediates (CH₃O*, HCHO, HCOOCH₃ and HCOO*) and reaction products (H₂ and CO₂) are captured after the introduction of CH₃OH/He on 4.25Cu/Cu(Al)O_x at 240 °C (Fig. 4c, red). In contrast, when CH₃OH + H₂O is co-introduced (Fig. 4b), the relative intensity of HCOOCH₃ decreases accompanied with the increase of HCOO* signal (Fig. 4c, blue). This indicates the co-introduction of H₂O greatly promotes the hydrolysis of HCOOCH₃. The same results are also found on 0.95Cu/Cu(Al)O_x and 7.18Cu/Cu(Al)O_x samples, but the higher relative intensities of HCOO* and CH₃O* suggest a lower reaction rate (Supplementary Fig. 35).”

● **Page 15, Line 20: rephrase:** “Furthermore, we performed the pulse experiments of HCOOCH₃ and HCOOCH₃ + H₂O over 4.25Cu/Cu(Al)O_x catalyst at different reaction temperatures, respectively (Fig. 4d). In the former case, as the temperature increases from 150 to 240 °C, the signals of HCOOCH₃ decline accompanied with the rise of H₂, CO₂, CH₃O* and HCOO* signals due to the dissociation of HCOOCH₃. In the latter case, the presence of H₂O greatly promotes the conversion of HCOOCH₃, with significantly weakened HCOOCH₃ but enhanced CH₃O* and HCOO* signals. Therefore, the whole MSR reaction mechanism over 4.25Cu/Cu(Al)O_x catalyst is shown in Fig. 4e: CH₃OH firstly undergoes dehydrogenation to form CH₃O* and HCHO species; then HCHO experiences dimerization or reacts with CH₃O* to generate HCOOCH₃; subsequently, HCOOCH₃ hydrolyzes to form HCOOH and CH₃O*, and CH₃O* re-participates in the catalytic cycle; finally, the decomposition of HCOOH occurs to produce CO₂ and H₂. In addition, according to the MS spectra results (Fig. 4c, Supplementary Fig. 35 and Supplementary Note 1), the signals of CH₃O* and HCOO* are much stronger than those

of HCHO and HCOOCH₃ during the MSR reaction, signifying that the conversion of CH₃O* and HCOO* is a kinetically slower process. According to the kinetic studies (Fig. 3f, g), the fracture of C–H bonds in CH₃O* and HCOO* intermediates is proved as the rate-determining step, and water molecule promotes the decomposition of HCOOCH₃ but does not participate directly in the cleavage of C–H bonds.”

● **Page 18, Line 8: rephrase:** “Furthermore, when exposing 4.25Cu/Cu(Al)O_x catalyst to CH₃OH/H₂O/He from 50 to 270 °C (Fig. 5e), the signals of $\delta_{\text{C-H}}$, ν_{HCOO} and $\nu_{\text{C=O}}$ increase gradually, accompanied with the appearance of gas CO₂ (2380–2307 cm⁻¹) and OH⁻ group (3390 and 3723 cm⁻¹) (Supplementary Fig. 36), in accordance with the pulse experiments results (Fig. 4b).”

● **Page 18, Line 13: rephrase:** “Subsequently, we carried out *in situ* FT-IR to study the adsorption and reaction behavior of HCOOCH₃ intermediate at 240 °C on the 4.25Cu/Cu(Al)O_x catalyst (Fig. 5c, green lines). After the introduction of HCOOCH₃, the bands attributed to C=O bond (1741 and 1768 cm⁻¹) in HCOOCH₃ and the ones assigned to C–H and COO⁻ group from adsorbed HCOOCH₃ (1348, 1449 and 1582 cm⁻¹) are observed. After switching to a saturated water vapor (Fig. 5c, red lines), the C=O bond disappears accompanied with the gradual decline of HCOO* and CH₃O* species, corresponding to the HCOOCH₃ hydrolysis and C–H bond cleavage in reaction intermediates.”

● **Page 22, Line 18: rephrase:** “In addition, we also studied the formaldehyde oxidation route (Supplementary Figs. 67–70) and HCOOCH₃* route (Fig. 6g and Supplementary Figs. 58–66) over the Cu/CuAlO₂ model to reveal the role of H₂O in the MSR reaction. Notably, for the HCOOCH₃* pathway, H₂O mainly helps in the hydrolysis of methyl formate to form CH₃O* and HCOO* (Supplementary Figs. 62–64) with a reactively low energy barrier (0.72 eV). In contrast, for the formaldehyde oxidation pathway (Supplementary Figs. 67–69), the hydroxyl species from H₂O dissociation oxidizes CH₂O to CH₂OOH, followed by CH₂OOH dehydrogenation to generate HCOO with an extremely high energy barrier of 4.20 eV. Thus, the methyl formate pathway is favorable on surface of Cu/CuAlO₂, and the role of water molecule is to hydrolyze methyl formate to produce formate, which is consistent with the experimental results.”

● Fig. 3g, 4a–d, 5c, 5e and 6g have been supplemented in the revised manuscript.

Fig. 3g Reaction order of CH₃OH and H₂O over 4.25Cu/Cu(Al)O_x catalyst at 240 °C.

Fig. 4 MS signals for the pulse experiments of **a** methanol and **b** methanol-water (1:2) over 4.25Cu/Cu(Al)O_x at 240 °C, respectively. **c** Relative intensity of the reaction intermediates normalized by CO₂ signal based on the results of **a** and **b**. **d** MS signals for the pulse experiments of methyl formate and methyl formate-water (1:2) over 4.25Cu/Cu(Al)O_x from 120 to 240 °C, respectively.

Fig. 5c *In situ* FT-IR spectra of 4.25Cu/Cu(Al)O_x along with the sequential introduction of HCOOCH₃/He (1–15 min) and H₂O/He (15–30 min).

Fig. 5e *In situ* FT-IR spectra of 4.25Cu/Cu(Al)O_x along with the introduction of CH₃OH/H₂O/He from 50 to 270 °C.

Fig. 6g Full potential reaction pathway of ESR reaction following the HCOOCH₃* mechanism over Cu/CuAlO₂ and Cu/Cu₂O, respectively. ‘TS’ denotes the transition state. Numbers located at the horizontal line represent free energy of corresponding intermediate (the inset gives the energy barrier of each transient state).

- Supplementary Figs. 67–70 have been supplemented in the revised Supplementary Information.

Supplementary Figure 67. Calculated potential energy diagram and corresponding geometric configurations for the oxidation of CH₂O on the surface of Cu(111)/CuAlO₂(101) (*, IS, TS and FS

represent the adsorption state, initial state, transition state and final state, respectively; E_a and ΔE is the energy barrier and thermodynamic energy).

Supplementary Figure 68. Calculated potential energy diagram and corresponding geometric configurations for the dehydrogenation of CH_2OOH on the surface of $\text{Cu}(111)/\text{CuAlO}_2(101)$ (*, IS, TS and FS represent the adsorption state, initial state, transition state and final state, respectively; E_a and ΔE is the energy barrier and thermodynamic energy).

Supplementary Figure 69. Calculated potential energy diagram and corresponding geometric configurations for the dehydrogenation of CH_2OO^* on the surface of $\text{Cu}(111)/\text{CuAlO}_2(101)$ (*, IS, TS and FS represent the adsorption state, initial state, transition state and final state, respectively; E_a and ΔE is the energy barrier and thermodynamic energy).

Supplementary Figure 70. Full-path analysis for MSR reaction following the formaldehyde oxidation mechanism over Cu/CuAlO₂. ‘TS’ denotes the transition state. Numbers located at the horizontal line represent the free energy of corresponding intermediates (the inset gives the energy barrier of each transient state).

(6) Beside these comments, there are other points, which need to be revised. Page 16, Blue and red shift of the IR frequencies over 0.95Cu/Cu(Al)O_x and 7.18Cu/Cu(Al)O_x samples. What it mean? Page 20. Dynamic reconstruction process of the Cu⁺-Cu⁰ interfacial sites during the whole MSR reaction. Please, what it mean.

Author reply: Thank you for this comment. As shown in the *in situ* FT-IR spectra, compared with 4.25Cu/Cu(Al)O_x (Fig. 5b), the absorption bands of CH₃O* and HCOO* intermediates show a blue-shift on 0.95Cu/Cu(Al)O_x (Supplementary Fig. 37b) whilst a red-shift on 7.18Cu/Cu(Al)O_x (Supplementary Fig. 38b), respectively. Normally, a higher frequency means a weaker adsorption strength of intermediates whilst a lower frequency indicates a stronger adsorption strength. Thus, the CH₃O* and HCOO* intermediates on 4.25Cu/Cu(Al)O_x display a moderate adsorption strength, which is favorable for the subsequent reaction. This is consistent with the DFT calculation results (Fig. 6a–c). The corresponding discussions have been improved in the revised manuscript.

In addition, the dynamic reconstruction process during the MSR reaction means that the geometric and electronic structure of Cu species at the Cu⁰-Cu⁺ interfacial sites undergo dynamic changes during

the MSR reaction. Concretely, as shown in the *operando* XAFS spectra (Fig. 5g–i and Supplementary Figs. 48–51), electron transfer occurs from the interfacial Cu sites to reaction intermediates after the introduction of CH₃OH, HCHO, CH₃COOH and H₂O, respectively. Correspondingly, both the coordination number and bond length of Cu–Cu bond decrease; whilst the coordination number and bond length of Cu⁺–O bond increase. After the reactive species is purged by He, the electron state and coordination structure of surface Cu species recover to its initial state, demonstrating that the Cu species undergoes an electronic and geometrical reconfiguration. These dynamic reconstructions are further supported by the Bader charge analysis and bond length variation from DFT calculation results (Supplementary Figs. 89–92). The corresponding discussions have been improved in the revised manuscript.

● **Page 21, Line 3: rephrase:** “*Operando* XAFS measurements were carried out to reveal the dynamic evolution in electronic structure and coordination state of Cu⁰–Cu⁺ synergistic sites during the MSR reaction processes. As shown in Fig. 5g, compared with the pristine 4.25Cu/Cu(Al)O_x sample (green line), the intensity of the white line peak increases along with the sequential introduction of CH₃OH (orange line) and H₂O (blue line), indicating that the Cu species undergoes an electronic reconfiguration due to the electron transfer from catalyst to reactive species. Correspondingly, both the coordination number and bond length of Cu–Cu decrease after the introduction of CH₃OH and H₂O, as demonstrated by the EXAFS spectra (Fig. 5h,i) and fitting results in R space of Cu k-edge (Supplementary Figs. 47,48 and Supplementary Table 4). In contrast, the coordination number and bond length of Cu–O increase simultaneously, owing to the formation of additional Cu⁰–H and Cu⁺–O bonds. After He purging (pink line) for 15 min to remove the surface adsorbates, both the XANES and EXAFS spectra of 4.25Cu/Cu(Al)O_x restore to their initial states. The results substantiate that the Cu⁰–Cu⁺ interfacial sites participate in the substrate activation and C–H bonds cleavage.”

● **Page 23, Line 9: rephrase:** “The optimal adsorption configurations and charge density difference (CDD) (Fig. 6a–c, Supplementary Figs. 80,81 and Supplementary Note 5) of CH₃O* and HCOO* are calculated, whose adsorption energies give the following order: Cu(111) (–5.23 and –5.12 eV) > Cu/Cu₂O (–3.37 and –4.47 eV) > Cu/CuAlO₂ (–2.84 and –3.09 eV) > Cu₂O (–2.41 and –2.39 eV) > CuAlO₂ (–2.40 and –1.54 eV). In the case of Cu(111)/CuAlO₂(101) system (Fig. 6c), the oxygen atom in CH₃O* is co-adsorbed at the Cu⁰–Cu⁺ interfacial sites; whilst for HCOO*, the two oxygen atoms

are adsorbed at interfacial Cu⁰ and Cu⁺ sites, respectively. These special oxygen-end bridge adsorption configurations at the Cu⁰–Cu⁺ interfacial sites of Cu/CuAlO₂ confer a moderate adsorption strength of both intermediates, in agreement with the *in situ* FT-IR results.”

• Fig. 6a–c has been supplemented in the revised manuscript.

Fig. 6a–c Optimized adsorption configurations of CH₃O* and HCOO* accompanied by the charge density difference (CDD) on the **a** Cu(111), **b** CuAlO₂(101) and **c** Cu(111)/CuAlO₂(101) models, respectively.

• Supplementary Figs. 80 and 81 have been supplemented in the revised Supplementary Information.

Supplementary Figure 80. Adsorption configuration of CH₃O* species on the surface of **a₁,a₂** Cu₂O(111) and **b₁,b₂** Cu(111)/Cu₂O(111) (white, gray, red and orange balls represent H, C, O and Cu atoms, respectively). Corresponding charge density difference (CDD) of CH₃O* adsorption on the surface of **a₃** Cu₂O(111) and **b₃** Cu(111)/Cu₂O(111).

Supplementary Figure 81. Adsorption configuration of HCOO* species on the surface of **a1,a2** Cu₂O(111) and **b1,b2** Cu(111)/Cu₂O(111) (white, gray, red and orange balls represent H, C, O and Cu atoms, respectively). Corresponding charge density difference (CDD) of HCOO* adsorption on the surface of **a3** Cu₂O(111) and **b3** Cu(111)/Cu₂O(111).

(7) Additional comments to the authors: DRIFT and Raman are not quantitative techniques. Therefore, it is not appropriate at all, to discuss differences in peak intensities among different samples, unless an internal reference is used for spectra normalization.

Author reply: Thank you for this comment. According to this comment, we removed the Raman data in the revised manuscript. However, for the *in situ* FT-IR data, we used a normalized method to perform a semi-quantitative analysis on band intensity vs. reaction time. Concretely, for the individual sample, the relative adsorption strength (CH₃O*/CD₃O* ratio and HCOO*/DCOO* ratio) after ventilating CH₃OH and CD₃OD at 15 min was used as the index to evaluate the consumption rates of CH₃O* (R_{H_m}) and HCOO* species (R_{H_f}) relative to CD₃O* (R_{D_m}) and DCOO* species (R_{D_f}) (Supplementary Fig. 42). A similar normalized method has also been used for semi-quantitative analysis of *in situ* DRIFT spectra in previous studies (*Nat. Catal.* 2022, 5, 99; *J. Catal.* 2021, 399, 121; *Nat. Catal.*, 2022, 5, 1038). The corresponding discussions have been improved in the revised manuscript and Supplementary Information.

● **Page 20, Line 10: rephrase:** “In addition, *in situ* FT-IR measurements for CD₃OD conversion were also performed to study the isotope effects of C–D bonds cleavage in CD₃O* and DCOO*

intermediates over the three catalysts (Supplementary Figs. 39–41). After switching to a saturated H₂O vapor, the consumption rates of CH₃O* (R_{H_m}) and HCOO* species (R_{H_f}) relative to CD₃O* (R_{D_m}) and DCOO* species (R_{D_f}) were calculated, where the R_{H_m} , R_{H_f} , R_{D_m} and R_{D_f} were the absolute value of slope from linear fitting (Supplementary Fig. 42 and Supplementary Note 2). The slight isotopic effect indicates that the C–H bonds breakage is significantly promoted on the surface of 4.25Cu/Cu(Al)O_x catalyst.”

• **Supplementary Information, Page 48, Line 2: rephrase:** “Supplementary Figs. 39–41 shows *in situ* FTIR spectra of 0.95Cu/Cu(Al)O_x, 4.25Cu/Cu(Al)O_x and 7.18Cu/Cu(Al)O_x catalysts along with adsorption of CD₃OD/He (1–15 min) and subsequent switching to H₂O/He (15–30 min) at 240 °C. For the adsorption of CD₃OD, due to the isotopic effect, the vibration frequency of C–D bonds moves towards a lower wavenumber relative to C–H bonds. The band at 2000–2300 cm⁻¹ is attributed to the stretching vibration of C–D bonds; the ones at 1268, 1325 and 1595 cm⁻¹ is assigned to the bending vibration of C–D bonds and DCOO* species, respectively. The peaks at 1110, 1130 and 1153 cm⁻¹ are related to the bending vibration of C–D bonds in CD₃O* group. Based on *in situ* FT-IR results (Fig. 5b and Supplementary Figs. 37–41), the correlation between normalized peak area of C–H bonds in CH₃O* and HCOO* as well as C–D bonds in CD₃O* and DCOO* versus ventilation time of saturated water vapor (within 15–30 min) was established, respectively (Supplementary Fig. 42). Compared with corresponding non-deuterium species, the relative consumption rates decrease from 0.041 (R_{H_m}) and 0.034 (R_{H_f}) to 0.024 (R_{D_m}) and 0.021 (R_{D_f}) in the presence of 0.95Cu/Cu(Al)O_x catalyst, respectively (Supplementary Fig. 42a,b). For the 7.18Cu/Cu(Al)O_x catalyst, the relative consumption rates decrease from 0.037 (R_{H_m}) and 0.018 (R_{H_f}) to 0.027 (R_{D_m}) and 0.009 (R_{D_f}), respectively (Supplementary Figs. 42e,f). Remarkably, in the case of 4.25Cu/Cu(Al)O_x catalyst, the values merely show a slight decrease from 0.058 (R_{H_m}) and 0.042 (R_{H_f}) to 0.055 (R_{D_m}) and 0.040 (R_{D_f}) (Supplementary Figs. 42c,d). The slight isotopic effect in the transformation of CD₃O* and DCOO* indicates that the C–D bonds breakage is significantly promoted on the surface of 4.25Cu/Cu(Al)O_x catalyst. The results agree well with the catalytic evaluations and kinetics studies.”

• Supplementary Fig. 42 has been supplemented in the revised Supplementary Information.

Supplementary Figure 42. Linear fitting results from normalized peak area of C–H and C–D bonds in **a,c,e** CH₃O*/CD₃O* ratio and **b,d,f** HCOO*/DCOO* ratio vs. ventilation time of water (within 15–30 min) for the sample of **a,b** 0.95Cu/Cu(Al)O_x, **c,d** 4.25Cu/Cu(Al)O_x and **e,f** 7.18Cu/Cu(Al)O_x. Data are obtained from Fig. 5b and Supplementary Figs. 44–48, respectively (the slope represents the relative consumption rate).

(8) High X-ray power in the XPS studies, (for instance the 300 W used in this work) may result in photo-reduction, specifically in case of copper species. In XPS studies, a reliable way to determine the oxidation state of copper species is analysing the auger parameter.

Author reply: Thank you for this comment. We are sorry for the mistake: the “300 W” should be “300 K”, which represents the test temperature (room temperature). We have corrected this issue in the

supplementary methods. According to this comment, for the analysis of oxidation state of copper species, we implemented *quasi-in situ* Cu LMM Auger spectra, and the results were shown in Fig. 1c.

- Fig. 1c has been supplemented in the revised manuscript.

Fig. 1c *Quasi-in situ* Cu LMM AES spectra of γ Cu/Cu(Al)O_x and control samples.

Reviewer #3

Comments:

The authors describe a catalyst for methanol reforming where they claim a specific Cu-Cu(I) site that is responsible for the good performance of the catalyst. This is interesting work, with a significant amount of characterization work. Any yet I do not agree that their study meet their stated criteria of “a detailed study via employing spatially and temporally-resolved operando characterization techniques, kinetic investigations as well as theoretical calculations is imperative to shed light on the intrinsic active sites...”.

(1) There seems to be a disconnect between the information learned from bulk characterization techniques (e.g. XRD, XAS) and the catalytically active sites on particles that are I don't understand the comment: “Notably, compared with Cu₂O standard sample, the 2θ of Cu₂O(111) at 36.8° in these samples shifts to a higher diffraction angle, close to the CuAlO₂(101) lattice plane (PDF#40-1037), which indicates the formation of Cu⁺ stabilized by amorphous alumina”

when the Rietveld refinement confirms the presence of bulk Cu₂O.

Author reply: Thank you for this comment. Although the XRD and XAS belong to the bulk structure characterization techniques, they can be used to evaluate the surface active sites to some extent when the sample particle size ranges in the nanometer scale. As shown in the XRD patterns (Fig. 1b), compared with the standard Cu₂O(111) lattice plane (36.4°), the 2θ of Cu₂O(111) for these samples shifts to a higher diffraction angle (36.8°), which is located between Cu₂O(111) and CuAlO₂(101). Meanwhile, the reflection intensity of Cu₂O phase declines along with the decrease of Al content, indicating that the amorphous alumina at the Cu-Al₂O₃ interface helps to stabilize the Cu₂O phase with the formation of a CuAlO₂-like structure.

For the XAS spectra, from the fitting results and wavelet transform (Supplementary Figs. 17 and Supplementary Table 2), the longer bond length of Cu–O in γCu/Cu(Al)O_x samples (1.84 Å) relative to Cu₂O standard (1.81 Å) is related to the formation of unique Cu–O–Al geometric coordination in the CuAlO₂-like structure. This is consistent with the DFT calculation results (Fig. 1g–i), in which the Cu⁺–O bond length in Cu(111)/CuAlO₂(101) (1.90 Å) is significantly longer than that in Cu₂O(111) (1.83 Å) and Cu(111)/Cu₂O(111) (1.83 Å). This deduction is visually demonstrated by EELS mapping images (Fig. 2g–l). Furthermore, we carried out N₂O-titration (Supplementary Fig. 33) and CO-TPD (Supplementary Fig. 34) measurements to quantify the surface concentrations of Cu⁰ and Cu⁺ as well as the Cu⁰–Cu⁺ interfacial perimeter, so as to provide additional evidence to study the intrinsic active sites.

- Fig. 1b has been improved in the revised manuscript.

Fig. 1b XRD patterns of γ -Cu/Cu(Al)O_x and control samples.

(2) I don't understand the TPR data. For example, 0.95CuAlO_x shows only a small amount of reduction < 220 °C. However, the Rietveld refinement of the sample 0.95Cu/Cu(Al)O_x shows 31% metallic Cu. How are these data reconciled? How was this Cu reduced?

Author reply: Thank you for this comment. H₂-TPR is a dynamic reduction process, whilst catalyst activation is a long-term and static reduction process. For example, the heating rate of H₂-TPR is 10 °C min⁻¹, and the fast-heating rate leads to a certain lag relationship between reduction degree and reduction temperature. In the catalyst activation process, a sufficient reduction time at 220 °C ensures a full reduction degree of catalyst to steady state.

(3) It appears that the 220 °C reduction results in a kinetically controlled reduction of the different types of Cu present in the sample. How did the catalytic performance vary with a reduction temperature, say 50 °C below/above this temperature?

Author reply: Thank you for this comment. According to this comment, we compared the catalytic performance of 4.25Cu/Cu(Al)O_x catalysts obtained at different pretreatment temperatures (170, 220, 250, 270 and 300 °C, respectively). With the increase of reduction temperature, both the CH₃OH conversion and H₂ production rate increase first and then decrease, and the 4.25Cu/Cu(Al)O_x sample reduced at 220 °C displays the optimal catalytic performance. The XRD patterns and XANES spectra

indicate that the 4.25Cu/Cu(Al)O_x sample reduced at 220 °C gives an appropriate proportion of Cu⁺ species. The corresponding discussions have been improved in the revised manuscript.

● **Page 11, Line 14: rephrase:** “In addition, the studies on reduction temperature from 170 to 300 °C (Supplementary Fig. 26) show that the 4.25Cu/Cu(Al)O_x sample reduced at 220 °C with an appropriate proportion of Cu⁺ species (Supplementary Fig. 27) exhibits the highest catalytic activity.”

● Supplementary Figs. 26 and 27 have been supplemented in the revised Supplementary Information

Supplementary Figure 26. a Methanol conversion and **b** H₂ production rate in the presence of 4.25CuAlO_x samples reduced at 170, 220, 250, 270, 300 °C, respectively. The 4.25Cu/Cu(Al)O_x-220 herein and 4.25Cu/Cu(Al)O_x in the manuscript refer to the same catalyst. Reaction conditions: catalyst (0.25 g) + SiO₂ (2.50 g); liquid feed of S/C = 2 at 0.040 mL min⁻¹; He carrier at 50.0 mL min⁻¹; time on stream: 1.0 h.

Supplementary Figure 27. **a** XRD patterns, **b** XANES spectra, **c** EXAFS spectra at k-space and **d** R-space of the 4.25Cu/Cu(Al)O_x catalysts reduced at 170, 220, 250, 270 and 300 °C, respectively. The 4.25Cu/Cu(Al)O_x-220 herein and 4.25Cu/Cu(Al)O_x in the manuscript refer to the same catalyst.

(4) The size if the Cu and Cu₂O particles is very large – from 7 to 9 nm. How can there be a unique Cu/Cu(I) site between such large particles?

Author reply: Thank you for this comment. The Cu/Cu(I) site is not derived from the direct contact between Cu particles and Cu₂O particles. In fact, as shown in Fig. 2 and Supplementary Fig. 26, the Cu⁰–Cu⁺ sites in this work come from the interweaved interface of metal Cu particle and its surrounding Cu₂O counterpart, in which Cu⁰ is located in the center of Cu particle whilst Cu₂O is located at the edge of Cu particle stabilized by amorphous alumina support. Similar interface structure of Cu⁰–Cu⁺ sites has also been reported in previous studies including Cu/CeO₂ (*ACS Catal.* 2022, 12, 1315), Cu/ZnO (*Appl. Catal. A: Gen.* 2021, 616, 118072), Cu/SiO₂ (*Angew. Chem. Int. Ed.* 2018, 130, 1854; *Chem. Sci.* 2019, 10, 2578; *ACS Catal.* 2020, 10, 14694; *Appl. Catal. B: Environ.* 2022, 325, 122329) and Cu/La₂O₃ (*Appl. Catal. B: Environ.* 2019, 251, 119).

(5) XAS methodology: The authors state: “transmission mode with a standard Lytle ion chamber”. A Lytle ion chamber is used for fluorescence XAS measurements so the statement is incorrect.

Author reply: Thank you for this comment. We are very sorry for this description mistake. We have corrected it in the revised Supplementary Information.

● **Supplementary Information, Page 2, Line 16: rephrase: “*In situ* XAFS spectra of Cu K-edge were obtained on transmission mode with a standard transmission ion chamber detector (Cu foil as reference) on the beamline 1W1B of the Beijing Synchrotron Radiation Facility (BSRF), Institute of High Energy Physics (IHEP), Chinese Academy of Sciences (CAS)”.**

(6) The authors state: “For the operando MSR reaction, based on the above pretreatment, a certain amount of CH₃OH was carefully evaporated into the *in situ* cell in He flow (30 mL min⁻¹) at 200 °C along with signal collection; afterwards, saturated water vapor was introduced into the *in situ* cell under similar conditions, for the collection of XAFS spectra.” This is insufficient detail to allow an interested researcher to duplicate their findings.

Author reply: Thank you for this comment. We have elaborated the *operando* XAFS test conditions in the revised Supplementary Information.

● **Supplementary Information, Page 6, Line 20: rephrase: “Typically, the powdered sample (30 mg) was pressed into a self-supporting wafer and carefully installed into a reaction microdevice equipped with polyimide windows. Afterwards, the sample was pre-reduced in a 25% H₂/Ar mixture gas (30 mL min⁻¹) at 220 °C for 1 h, followed by cooling down to 200 °C in a high-purity He stream (30 mL min⁻¹) for the collection of initial XAFS spectrum. Subsequently, saturated methanol/formaldehyde/methyl formate/formic acid steam (30 °C) carried by He (30 mL min⁻¹) was carefully evaporated into the *in situ* cell at 200 °C to collect XAFS signals after 5 min; afterwards, saturated water vapor (30 °C) carried by He (30 mL min⁻¹) was introduced into the *in situ* cell at 200 °C for the collection of XAFS spectra after 5 min. Finally, the gas flow was switched to a pure He (30 mL min⁻¹) to purge the catalyst surface at 200 °C for 10 min and then XAFS spectra were collected.”**

(7) The authors state: “All the XAFS data were processed using Athena software package”. This is simply insufficient information as to how the XAS data were processed. What k-range was used for the FT, what R-range was used in the fitting, what was the value of S02 – and how was it determined.

Author reply: Thank you. This has been supplemented in the revised Supplementary Information.

• **Supplementary Information, Page 7, Line 7: rephrase:** “All the XAFS data were processed using Athena software package³, and the data ranges used for data fitting in k-range, R-range and S02 value are 2.5–12.0 Å⁻¹, 1.0–3.0 Å and 0.8, respectively.”

(8) The analysis of the XAS data is rudimentary at best, and must be more extensive for Nature Commun. Stating that the average valence state is between 0 and 1 conveys no useful information – that is know from the XPS and XRD, for example. At a minimum LCF analysis of the XANES data is needed.

Author reply: Thank you for this comment. According to this comment, we carried out LCF analysis of the XANES data and the results were shown in Supplementary Fig. 15. The average valence state of Cu particle (Cu_{AVS}) displays the following sequence: $0.95Cu/Cu(Al)O_x (+0.86) > 2.32Cu/Cu(Al)O_x (+0.56) > 3.06Cu/Cu(Al)O_x (+0.49) > 4.25Cu/Cu(Al)O_x (+0.42) > 5.27Cu/Cu(Al)O_x (+0.34) > 7.18Cu/Cu(Al)O_x (+0.25) > Cu/Al_2O_3 (+0.12)$. Corresponding discussion has been added in the revised manuscript.

• **Page 8, Line 1: rephrase:** “For the $yCu/Cu(Al)O_x$ samples, the intensity of white line peaks decreases gradually from $0.95Cu/Cu(Al)O_x$ to $7.18Cu/Cu(Al)O_x$, indicative of a decline in average valence state of Cu species (Cu_{AVS}). This is consistent with the results from linear combination fitting (LCF) analysis (Supplementary Fig. 15), where the Cu_{AVS} gradually decreases from +0.86 to +0.25 from $0.95Cu/Cu(Al)O_x$ to $7.18Cu/Cu(Al)O_x$ sample. In contrast, the control sample of Cu/Al_2O_3 displays the lowest Cu_{AVS} (+0.12).”

• Supplementary Fig. 15 has been supplemented in the revised Supplementary Information.

Supplementary Figure 15. Linear combination fitting (LCF) analysis on the XANES data of **a** $0.95\text{Cu}/\text{Cu}(\text{Al})\text{O}_x$, **b** $2.32\text{Cu}/\text{Cu}(\text{Al})\text{O}_x$, **c** $3.06\text{Cu}/\text{Cu}(\text{Al})\text{O}_x$, **d** $4.25\text{Cu}/\text{Cu}(\text{Al})\text{O}_x$, **e** $5.27\text{Cu}/\text{Cu}(\text{Al})\text{O}_x$, **f** $7.18\text{Cu}/\text{Cu}(\text{Al})\text{O}_x$ and **g** $4.20\text{Cu}/\text{Al}_2\text{O}_3$, respectively. Cu_{AVS} is the average valence state of Cu species.

(9) The EXAFS modeling of the Cu_2O has 2 O at 1.81Å and 12 Cu at 3Å – yet these are not the values reported in Table S2.

Author reply: Thank you for this comment. We have corrected this issue in Supplementary Table 2.

(10) The authors state: “the longer bond length of Cu–O in γ Cu/Cu(Al)O_x samples (1.84 Å) relative to Cu₂O standard (1.81 Å) implicates a distorted tetrahedral structure due to the partial substitution of Cu by Al, which is possibly related to the formation of unique Cu–O–Al geometric coordination”. There is absolutely no justification for this in the data presented in the manuscript. Table S2 contains no error bars on the data so how can the authors claim that it is possible to claim that 1.81 and 1.84 Å are statistically different?

Author reply: Thank you for this comment. The error bars have been added in Supplementary Table 2, in which the fitting error was within 0.02 Å. In addition, we further evaluated the Cu–O bond length *vis* DFT calculations (Fig. 1g–i). The Cu–O bond in Cu(111)/CuAlO₂(101) (1.90 Å) is significantly longer than that in Cu₂O(111) (1.83 Å) and Cu(111)/Cu₂O(111) (1.83 Å), which is related to the unique Cu–O–Al geometric coordination from CuAlO₂-like structure. It should be noted that the smaller difference in Cu–O bond length in EXAFS is associated with the effect of bulk phase averaging.

- Fig. 1g–i has been supplemented in the revised manuscript.

Fig. 1 Cu–O bond length in **g** Cu₂O(111), **h** Cu(111)/Cu₂O(111) and **i** Cu(111)/CuAlO₂(101) systems based on DFT calculations (red: O; orange: Cu; purple: Al).

(11) Moreover: “Notably, the coordination number of Cu–Cu bond in γ Cu/Cu(Al)O_x increases from 5.1 to 8.2 whilst that of Cu–O bond declines from 1.7 to 0.5 along with the increment of Cu/Al ratio, in accordance with the valence state distribution of Cu species in these samples”.

How are the CN's from the XAS reconciled with the fractions of Cu and Cu₂O determined from the Rietveld refinement? There is no such thing as *quasi-in situ* XPS.

Author reply: Thank you for this comment. According to this comment, we performed a normalization treatment (*Phys. Chem. Chem. Phys.* 2010, 12, 5562) to correlate the coordination number of Cu–O and Cu–Cu bonds with the Cu₂O/Cu ratio for these samples. The relevant calculation method was added in the revised Supplementary Information. Remarkably, the Cu₂O/Cu ratios from XRD Rietveld refinement and EXAFS analysis exhibit a similar change trend (Supplementary Fig. 18). The results are in accordance with the valence distribution of Cu species obtained from *quasi-in situ* XPS, CO-DRIFT and XAFS-LCF analysis for these samples. As shown in Supplementary Fig. 18, the ratios of surface Cu⁺/Cu⁰ and bulk Cu₂O/Cu decrease along with the increment of Cu/Al molar ratio. Corresponding details and discussions have been added in the revised manuscript.

● **Supplementary Information, Page 4, Line 19: rephrase:** “Cu₂O/Cu fraction from EXAFS spectra. A normalization treatment was performed to correlate the coordination number (CN) of Cu–O and Cu–Cu bonds with the Cu₂O/Cu fraction for these samples. The calculation equations are as follows: $X_{\text{Cu}_2\text{O}} = \text{CN}_{\text{Cu-O}}/\text{CN}_{\text{Cu}_2\text{O}}/(\text{CN}_{\text{Cu-O}}/\text{CN}_{\text{Cu}_2\text{O}} + \text{CN}_{\text{Cu-Cu}}/\text{CN}_{\text{Cu}})$; $X_{\text{Cu}} = \text{CN}_{\text{Cu-Cu}}/\text{CN}_{\text{Cu}}/(\text{CN}_{\text{Cu-O}}/\text{CN}_{\text{Cu}_2\text{O}} + \text{CN}_{\text{Cu-Cu}}/\text{CN}_{\text{Cu}})$; $\gamma = X_{\text{Cu}_2\text{O}}/X_{\text{Cu}}$, where the CN_{Cu₂O} and CN_{Cu} are 2 and 12, corresponding to the CNs in the first shell of Cu–O and Cu–Cu bonds in Cu₂O and Cu standard samples, respectively. The CN_{Cu–O} and CN_{Cu–Cu} are the CNs of Cu–O and Cu–Cu bonds obtained from EXAFS (Supplementary Table 2), and γ is the ratio of Cu₂O/Cu.”

● **Page 8, Line 19: rephrase:** “We further correlated the coordination number of Cu–O and Cu–Cu bonds with the fraction of Cu₂O/Cu in these samples. Based on the XRD Rietveld refinement, *quasi-in situ* Cu LMM, *in situ* CO-DRIFTS, XAFS-LCF and EXAFS-Fit analysis results, a negative correlation between Cu⁺/Cu⁰ ratio and Cu/Al ratio is established, demonstrating the significant role of amorphous alumina in stabilizing Cu⁺ species (Supplementary Fig. 18).”

● Supplementary Fig. 18 has been supplemented in the revised Supplementary Information.

Supplementary Figure 18. Ratios of Cu^+/Cu^0 obtained from XRD Rietveld refinement, *quasi-in situ* Cu LMM, CO-DRIFT, XAFS-LCF and EXAFS-Fit analysis results as a function of Cu/Al molar ratio based on ICP-AES for these six $y\text{Cu}/\text{Cu}(\text{Al})\text{O}_x$ samples.

REVIEWER COMMENTS

Reviewer #1 (Remarks to the Author):

Many of the author's answers to my previous comments are satisfactory, but I have no clear their comments involving the new figure 3g. In methanol steam reforming, the ratio of methanol to water is 1:1. What is the pressure of water when the methanol pressure is changed? Also, what is the pressure of methanol when the pressure of water is changed? How the variation with the pressure of methanol implies that the rate determining step is the cleavage of C-H bonds?

In Figure 5d, the introduction of water leads to a reduction in the concentration of oxygenate intermediates present in the surface of the catalyst. This is consistent with the comments in my previous review. Do the DFT results in Figure 6 contradict the experimental data in Figure 5d?

These are key issues that need to be addressed before the article is accepted for publication.

Reviewer #2 (Remarks to the Author):

the authors have responded carefully to all the comments done by the reviewers and are convincing. I consider the manuscript suitable for publication.

Reviewer #3 (Remarks to the Author):

The authors have made wonderful effort to address the concerns of the reviewers including conducting additional experiments, characterization, and calculations and they should be commended for this effort. However, there are still some aspects that have not been addressed and need further clarification.

I must admit that even with all of the exemplary characterization data in the manuscript the actual structure that they propose is not clear to me. This was also true of another reviewer. In the response the authors again describe in words that are still not 100% clear. ("the CuO-Cu⁺ sites in this work come from the interweaved interface of metal Cu particle and its surrounding Cu₂O counterpart, in which CuO is located in the center of Cu particle whilst Cu₂O is located at the edge of Cu particle stabilized by amorphous alumina support"). What is an interweaved interface? I suggest a schematic cartoon of the Cu nanoparticle on the Cu₂O stabilized on the Cu-doped alumina surface, with the size of the Cu NP increasing with wt% Cu, with the unique interface sites indicated, would go a long way to making it visually clear to the reader, and certainly to me.

The authors state that the longer bond length of Cu-O in γ Cu/Cu(Al)Ox (1.84 Å) samples relative to Cu₂O

standard (1.81 Å) based on the EXAFS spectra is proof of the Cu-O-Al interaction. I challenge this conclusion. This is a 0.03 Å difference, and the error bars of this bond length are ± 0.02 Å. If the authors insist that the 0.01 Å difference is real, then they need to provide additional justification. I do note that in the 92 figures in the SI the authors do not show the fits to the EXAFS data. These fits must be shown as a minimum. This result is not “verified” by the DFT – the DFT can be used to support their interpretation, but it does not prove it is correct.

The authors state: “the Cu⁰-Cu⁺ sites in this work come from the interweaved interface of metal Cu particle and its surrounding Cu₂O counterpart, in which Cu⁰ is located in the center of Cu particle whilst Cu₂O is located at the edge of Cu particle stabilized by amorphous alumina support”. So from the oxidic copper perspective does this imply that there is Cu-O-Al from the interface and Cu-O from the Cu₂O? If so, then this leads to even less credence for the 0.01 Å difference in the Cu-O bond length as there are multiple bond lengths.

Related to this is their normalization treatment to correlate the coordination number (CN) of Cu-O and Cu-Cu bonds with the Cu₂O/Cu fraction for these samples. In this method, they use the following formula $X_{Cu_2O} = \frac{CN_{Cu-O}}{CN_{Cu_2O}} / (\frac{CN_{Cu-O}}{CN_{Cu_2O}} + \frac{CN_{Cu-Cu}}{CN_{Cu}})$ which is a derivation of a method by Beale et al. for determining coordination number for mixed phases. If I understand correctly in the authors case there is no mixed phase, and thus it is simply $X_{Cu_2O} = \frac{CN_{Cu-O}}{CN_{Cu_2O}}$. But even here there is inconsistency. Above the authors use the justification of the increased bond length which “demonstrates a distorted tetrahedral structure owing to the interfacial Cu-O-Al coordination (CuAlO₂-like structure), and thus the normalization would be divided by the structure of CuAlO₂ and not Cu₂O.

Response to Reviewers

Reviewer #1

Comments:

(1) Many of the author's answers to my previous comments are satisfactory, but I have no clear their comments involving the new figure 3g. In methanol steam reforming, the ratio of methanol to water is 1:1. What is the pressure of water when the methanol pressure is changed? Also, what is the pressure of methanol when the pressure of water is changed? How the variation with the pressure of methanol implies that the rate determining step is the cleavage of C-H bonds?

Author reply: Thank you for this comment. For the reaction order measurements of CH₃OH and H₂O in Fig. 3g, we changed the molar ratio of water to methanol as well as the flow rate of carrier gas to maintain a fixed partial pressure of one substrate and adjust the partial pressure of another. The corresponding experimental detail has been added in the Supplementary Information.

With respect to the reaction kinetics, based on the data fitting (Fig. 3g), the reaction rate of CH₃OH shows a more significant concentration-dependence relationship (0.485) than that of H₂O (0.013), indicating that the activation of CH₃OH is involved in the key step. In addition, according to the KIE isotope test results (Fig. 3f), CD₃OD gives a more obvious KIE effect (3.84–6.56) than CH₃OD (0.95–1.06), which demonstrates that the cleavage of C–H bond in methanol is the rate-determining step of MSR reaction. DFT calculation results further verify this issue.

• Supplementary Information, Page 7, Line 11: rephrase:

“**Reaction order measurement.** We changed the molar ratio of water to methanol as well as the flow rate of carrier gas to maintain a fixed partial pressure of one substrate and adjust the partial pressure of another. For the measurement of water reaction order, we performed 5 sets of water:methanol ratios (1:0.5, 1:0.75, 1:1, 1:1.5 and 1:2, respectively) with a constant flow velocity of methanol solution (0.04 mL min⁻¹), and the flow rate of carrier gas was 50.0, 63.3,

81.7, 91.5 and 104.3 mL min⁻¹, respectively. Under the above five sets of reaction conditions, the partial pressure of methanol vapor is approximately a fixed value (13.9 kPa), and the partial pressure of water vapor is 27.9, 20.8, 13.9, 10.4 and 6.9 kPa, respectively, calculated based on the gas state equation. Similarly, for the measurement of methanol reaction order, the water:methanol ratio and flow velocity of methanol solution are consistent with the above situation, and the flow rate of carrier gas is 50.0, 35.1, 25.0, 12.9 and 5.7 mL min⁻¹, respectively. The partial pressure of water vapor is approximately a fixed value (27.8 kPa), and the partial pressure of methanol vapor is 13.9, 20.9, 27.9, 41.8 and 55.8 kPa, respectively.”

(2) In Figure 5d, the introduction of water leads to a reduction in the concentration of oxygenate intermediates present in the surface of the catalyst. This is consistent with the comments in my previous review.

Author reply: Thank you for this comment. Indeed, as you mentioned, the introduction of water leads to a reduction of oxygenate intermediates on the catalyst surface. On the other hand, we cannot deny that H₂O imposes competitive adsorption with the intermediate, but this effect is much less than the influence of H₂O on the oxygenate intermediates. As demonstrated by the *operando* pulse experiments (Fig. 4), the introduction of water greatly promotes a rapid conversion of intermediates (methoxy and methyl formate) to formate and CO₂.

(3) Do the DFT results in Figure 6 contradict the experimental data in Figure 5d? These are key issues that need to be addressed before the article is accepted for publication.

Author reply: Thank you for this comment. According to this suggestion and the results from Figure 5d, we recalculated the conversion path from formaldehyde dimerization (2CH₂O* → CH₂OOCH₂*) and CH₂OOCH₂* direct dehydrogenation process to methyl formate (CH₂OOCH₂* → HCOOCH₃*) on the surface of Cu/CuAlO₂ and Cu/Cu₂O models, respectively, and compared this route with the previously calculated hydroxyl assisted dehydrogenation route (CH₂OOCH₃* + OH* → HCOOCH₃* + H₂O*). The corresponding discussion has been added in the revised Supplementary Information and manuscript.

• **Supplementary Information, Page 90, Line 6: rephrase:**

“**Supplementary Note 6.** According to the calculation result, the formaldehyde dimerization ($2\text{CH}_2\text{O}^* \rightarrow \text{CH}_2\text{OOCH}_2^*$) is a spontaneous barrier free process on both Cu/CuAlO₂ and Cu/Cu₂O models. For the formation of methyl formate intermediate, compared with the hydroxyl assisted dehydrogenation route (1.83 eV on Cu/CuAlO₂ and 1.61 eV on Cu/Cu₂O: $\text{CH}_2\text{OOCH}_3^* + \text{OH}^* \rightarrow \text{HCOOCH}_3^* + \text{H}_2\text{O}^*$) (Supplementary Figs. 65,80), the conversion of $\text{CH}_2\text{OOCH}_2^*$ to methyl formate through direct dehydrogenation process (1.06 eV on Cu/CuAlO₂ and 1.18 eV on Cu/Cu₂O: $\text{CH}_2\text{OOCH}_2^* \rightarrow \text{HCOOCH}_3^*$) (Supplementary Figs. 66,67,81,82) shows a lower reaction energy barrier, indicating that the conversion of formaldehyde to methyl formate is the favorable path for methyl formate formation ($2\text{CH}_2\text{O}^* \rightarrow \text{HCOOCH}_3^*$). Subsequently, the introduction of water promotes the hydrolysis of methyl formate (0.72 eV on Cu/CuAlO₂ and 1.15 eV on Cu/Cu₂O: $\text{HCOOCH}_3^* + \text{OH}^* \rightarrow \text{HCOOH}^* + \text{CH}_3\text{O}^*$) (Supplementary Figs. 69,70,84,85). The calculation results are in good agreement with the experimental data in Fig. 5d.”

• **Page 23, Line 12: rephrase:** “The Gibbs free energy diagrams and detailed schematic representation of the stepwise reaction for HCOOCH₃^{*} route on Cu/Cu₂O and Cu/CuAlO₂ are displayed in Fig. 6g. Distinctly, the activation barriers for CH₃O^{*} formation (1.49 eV on Cu/CuAlO₂ and 1.38 eV on Cu/Cu₂O: $\text{CH}_3\text{OH}^* \rightarrow \text{CH}_3\text{O}^* + \text{H}^*$) (Supplementary Figs. 62,77), H₂O dissociation (0.56 eV on Cu/CuAlO₂ and 0.88 eV on Cu/Cu₂O: $\text{H}_2\text{O}^* \rightarrow \text{OH}^* + \text{H}^*$) (Supplementary Figs. 64,68,79,83), HCOOCH₃^{*} formation (1.06 eV on Cu/CuAlO₂ and 1.18 eV on Cu/Cu₂O: $2\text{CH}_2\text{O}^* \rightarrow \text{HCOOCH}_3^*$) (Supplementary Figs. 66,67,81,82 and Supplementary Note 6) and HCOOCH₃^{*} hydrolysis (0.72 eV on Cu/CuAlO₂ and 1.15 eV on Cu/Cu₂O: $\text{HCOOCH}_3^* + \text{OH}^* \rightarrow \text{HCOOH}^* + \text{CH}_3\text{O}^*$) (Supplementary Figs. 69,70,84,85) are lower than the dehydrogenation reactions of CH₃O^{*} (2.69 eV on Cu/CuAlO₂ and 2.85 eV on Cu/Cu₂O: $\text{CH}_3\text{O}^* \rightarrow \text{CH}_2\text{O}^*$) (Supplementary Figs. 63,78) and HCOO^{*} (2.13 eV on Cu/CuAlO₂ and 2.25 eV on Cu/Cu₂O: $\text{HCOO}^* \rightarrow \text{CO}_2^* + \text{H}^*$) (Supplementary Figs. 72,87), indicating that the cleavage of C–H bonds in CH₃O^{*} and HCOO^{*} acts as the rate-determining step. This is consistent with the experimental results.”

• Supplementary Figs. 66, 67, 81 and 82 have been supplemented in the revised Supplementary Information.

Supplementary Figure 66. Calculated potential energy diagram and corresponding geometric structures for the dehydrogenation of H_2COOCH_2 to HCOOCH_2 on the surface of $\text{Cu}(111)/\text{CuAlO}_2(101)$ (*, IS, TS and FS represent the adsorption state, initial state, transition state and final state, respectively. E_a and ΔE are the energy barrier and thermodynamic energy).

Supplementary Figure 67. Calculated potential energy diagram and corresponding geometric structures for the hydrogenation of HCOOCH_2 to HCOOCH_3 on the surface of $\text{Cu}(111)/\text{CuAlO}_2(101)$ (*, IS, TS and FS represent the adsorption state, initial state, transition state and final state, respectively. E_a and ΔE are the energy barrier and thermodynamic energy).

Supplementary Figure 81. Calculated potential energy diagram and corresponding geometric structures for the dehydrogenation of H_2COOCH_2 to HCOOCH_2 on the surface of $\text{Cu}(111)/\text{Cu}_2\text{O}(111)$ (*, IS, TS and FS represent the adsorption state, initial state, transition state and final state, respectively. E_a and ΔE are the energy barrier and thermodynamic energy).

Supplementary Figure 82. Calculated potential energy diagram and corresponding geometric structures for the hydrogenation of HCOOCH_2 to HCOOCH_3 on the surface of $\text{Cu}(111)/\text{Cu}_2\text{O}(111)$ (*, IS, TS and FS represent the adsorption state, initial state, transition state and final state, respectively. E_a and ΔE are the energy barrier and thermodynamic energy).

- Fig. 6g has been improved in the revised manuscript.

Fig. 6g Full potential reaction pathway of ESR reaction following the HCOOCH_3^* mechanism on the surface of Cu/CuAlO_2 and $\text{Cu/Cu}_2\text{O}$, respectively. ‘TS’ denotes the transition state. Numbers located at the horizontal line represent free energy of corresponding intermediate, and the inset gives the energy barrier of each transient state.

Reviewer #2

Comments:

The authors have responded carefully to all the comments done by the reviewers and are convincing. I consider the manuscript suitable for publication.

Author reply: Thank you very much for this comment.

Reviewer #3

Comments:

The authors have made wonderful effort to address the concerns of the reviewers including conducting additional experiments, characterization, and calculations and they should be commended for this effort. However, there are still some aspects that have not been addressed and need further clarification.

(1) I must admit that even with all of the exemplary characterization data in the manuscript the actual structure that they propose is not clear to me. This was also true of another reviewer. In the response the authors again describe in words that are still not 100% clear. (“the Cu⁰–Cu⁺ sites in this work come from the interweaved interface of metal Cu particle and its surrounding Cu₂O counterpart, in which Cu⁰ is located in the center of Cu particle whilst Cu₂O is located at the edge of Cu particle stabilized by amorphous alumina support”). What is an interweaved interface? I suggest a schematic cartoon of the Cu nanoparticle on the Cu₂O stabilized on the Cu-doped alumina surface, with the size of the Cu NP increasing with wt% Cu, with the unique interface sites indicated, would go a long way to making it visually clear to the reader, and certainly to me.

Author reply: Thank you for this comment. According to this suggestion, we added a schematic diagram in the revised Supplementary Information to help readers understand the catalyst structure. The corresponding discussion has been added in the revised manuscript and Supplementary Information.

• **Page 10, Line 19: rephrase:** “Thus, the 4.25Cu/Cu(Al)O_x sample is featured by aluminum-stabilized Cu⁺ adjacent to Cu⁰ nanoparticle immobilized on Al₂O₃ support, whose schematic structure diagram is shown in Supplementary Fig. 28 (Supplementary Note 1).”

• **Supplementary Information, Page 34, Line 5: rephrase:**

“**Supplementary Note 1.** As shown in Supplementary Fig. 28, both the size of Cu nanoparticles and the Cu/Al ratio increase with the increment of Cu loading. The Cu⁰ site is located at the exterior surface of Cu nanoparticles whilst Cu⁺ is located at the edge of Cu particles that has a strong interaction with Al₂O₃ support, resulting in the formation of specific Cu⁰–Cu⁺ interface sites as demonstrated by EELS results. XRD and EXAFS verify that such Cu⁺ species exists as Cu₂O with partial doping of aluminum. However, this Cu⁺ species is difficult to be reduced to a metallic state due to the stabilizing effect of amorphous alumina on Cu₂O, which becomes more significant with the increase of Al content as confirmed by H₂-TPR results.”

- Supplementary Fig. 28 has been supplemented in the revised Supplementary Information.

Supplementary Figure 28. Schematic structure diagram of the Cu/Cu(Al)O_x samples with increasing Cu loading (from **a** to **c**). The purple region indicates the Cu⁰–Cu⁺ interface sites. Red, earthy yellow, green and blue balls denote O, Cu⁺, Cu⁰ and Al, respectively.

(2) The authors state that the longer bond length of Cu–O in yCu/Cu(Al)O_x (1.84 Å) samples relative to Cu₂O standard (1.81 Å) based on the EXAFS spectra is proof of the Cu–O–Al interaction. I challenge this conclusion. This is a 0.03 Å difference, and the error bars of this bond length are ± 0.02 Å. If the authors insist that the 0.01 Å difference is real, then they need to provide additional justification. I do note that in the 92 figures in the SI the authors do not show the fits to the EXAFS data. These fits must be shown as a minimum. This result is not “verified” by the DFT – the DFT can be used to support their interpretation, but it does not prove it is correct. The authors state: “the Cu⁰–Cu⁺ sites in this work come from the interweaved interface of metal Cu particle and its surrounding Cu₂O counterpart, in which Cu⁰ is located in the center of Cu particle whilst Cu₂O is located at the edge of Cu particle stabilized by amorphous alumina support”. So from the oxidic copper perspective does this imply that there is Cu–O–Al from the interface and Cu–O from the Cu₂O? If so, then this leads to even less credence for the 0.01 Å difference in the Cu–O bond length as there are multiple bond lengths.

Author reply: Thank you for this comment. Actually, EXAFS technique is a kind of characterization that displays the bulk phase information with an average effect. In addition, as the reviewer mentioned, there are two types of Cu–O bond contributors (Cu–O–Cu and Cu–O–Al) in the sample of Cu/Cu(Al)O_x, and therefore we cannot obtain precise coordination information of Cu–O–Al from such characterization. This results in a rather small difference in

the average Cu–O bond length, which is a disadvantage of EXAFS characterizations. Nevertheless, similar comparative studies have been reported in previous papers. For example, a small difference in bond length (0.01–0.02 Å) through EXAFS was used as primary evidence to study the tensile or compressive strain of chemical bonds by Li et al (*J. Am. Chem. Soc.* 2022, 144, 19619; *Adv. Mater.* 2019, 31, 1903616).

To further prove the above conclusion, 4.25CuAlO_x catalyst precursor was calcined at various temperatures (500–800 °C) to regulate the metal-support interaction (Cu–Al₂O₃) and the doping degree of amorphous alumina on Cu₂O. In addition, we built 14 modified Cu₂O models in which Cu⁺ is substituted by Al³⁺ with various replacement percentages in order to reveal the doping effect of amorphous alumina on Cu₂O, and calculated the average bond length of Cu–O bond (Supplementary Fig. 21). The relevant content has been added in the revised manuscript and Supplementary Information.

• **Page 9, Line 3: rephrase:** “Furthermore, we changed the calcination temperature of 4.25CuAlO_x precursor from 500 to 800 °C to regulate the doping degree of amorphous alumina on Cu₂O. As shown in the H₂-TPR curves (Supplementary Fig. 19a), the reduction peak moves towards higher temperature with the increase of precursor roasting temperature, signifying an enhanced Cu–Al₂O₃ interaction. After the subsequent hydrogen activation treatment, XRD patterns show that from 4.25Cu/Cu(Al)O_x-500 to 4.25Cu/Cu(Al)O_x-800 sample (Supplementary Fig. 19b,c), the Cu₂O(111) reflection shifts to higher 2θ direction accompanied with increased peak intensity, which indicates a decreased Cu₂O cell volume due to the doping of Al atoms with smaller radius. Moreover, XPS (Supplementary Fig. 19d) and XAFS (Supplementary Figs. 19e,f, 20 and Supplementary Table 3) results demonstrate that the Cu⁺/Cu⁰ ratio (from 0.59 to 1.26) and the average bond length of Cu–O (from 1.84 to 1.89 Å) increase in sequence from 4.25Cu/Cu(Al)O_x-500 to 4.25Cu/Cu(Al)O_x-800, which is related to an enhanced proportion of Cu–O–Al geometric coordination. In addition, we built 14 modified Cu₂O models in which Cu⁺ is substituted by Al³⁺ with various replacement percentages (Supplementary Fig. 21a–n). The calculated average bond length of Cu–O increases from 1.83 to 1.87 Å as the atom ratio of Al/Cu increases from 0 to 7.7% (Supplementary Fig. 21o), which is consistent with the experiment results.”

- Supplementary Figs. 19, 20 and 21 have been supplemented in the revised Supplementary Information.

Supplementary Figure 19. **a** H_2 -TPR curves, **b,c** XRD patterns, **d** *quasi-in situ* Cu LMM AES spectra, **e** Cu K-edge XANES and **f** Cu K-edge EXAFS spectra of $4.25Cu/Cu(Al)O_{x-y}$ samples ($y = 500, 600, 700, 800$ °C, which denotes the roasting temperature of precursor. The $4.25Cu/Cu(Al)O_{x-500}$ herein refers to the sample of $4.25Cu/Cu(Al)O_x$ in the manuscript).

Supplementary Figure 20. Fitting results of EXAFS spectra at Cu K-edge of **a** $4.25Cu/Cu(Al)O_{x-600}$, **b** $4.25Cu/Cu(Al)O_{x-700}$ and **c** $4.25Cu/Cu(Al)O_{x-800}$ sample (the black line: experimental data; the red and blue lines: fitting curves).

Supplementary Figure 21. Modified Cu₂O models with substitution of Cu⁺ by Al³⁺ with various replacement percentage: **a,b** 0% (96 Cu and 48 O atoms), **c,d** 1.1% (93 Cu, 1 Al and 48 O atoms), **e,f** 2.2% (90 Cu, 2 Al and 48 O atoms), **g,h** 3.4% (87 Cu, 3 Al and 48 O atoms), **i,j** 4.8% (84 Cu, 4 Al and 48 O atoms), **k,l** 6.2% (81 Cu, 5 Al and 48 O atoms), **m,n** 7.7% (78 Cu, 6 Al and 48 O atoms). **o** Average bond length of Cu–O from 30 data points as a function of replacement percentage of Al to Cu.

• Supplementary Table 3 has been supplemented in the revised Supplementary Information.

Supplementary Table 3. EXAFS fitting parameters at the Cu K-edge for various samples

Sample	Shell	R (Å) ^a	CN ^b	σ^2 (10^{-3}\AA^2) ^c	ΔE_0 (eV) ^d	R factor (%) ^e
--------	-------	----------------------	-----------------	---	--------------------------------	-----------------------------

Cu-foil	Cu–Cu	2.54(0.01)	12.0(0.2)	8.5	4.52	0.2
Cu₂O	Cu–O	1.81(0.01)	2.0(0.1)	3.7	8.17	2.0
	Cu–Cu	3.01(0.02)	12.0(0.1)	3.4		
4.25Cu/Cu(Al)O_x-500	Cu–O	1.84(0.01)	0.9(0.2)	9.1	-3.57	0.8
	Cu–Cu	2.54(0.01)	6.9(0.2)	9.1		
4.25Cu/Cu(Al)O_x-600	Cu–O	1.85(0.01)	1.1(0.2)	7.5	4.99	1.3
	Cu–Cu	2.54(0.01)	6.4(0.1)	8.8		
4.25Cu/Cu(Al)O_x-700	Cu–O	1.87(0.02)	1.4(0.2)	7.6	6.31	1.6
	Cu–Cu	2.54(0.01)	5.8(0.3)	9.1		
4.25Cu/Cu(Al)O_x-800	Cu–O	1.89(0.01)	1.6(0.1)	8.8	5.06	1.5
	Cu–Cu	2.55(0.02)	5.5(0.2)	9.3		

^a Bond length.

^b Coordination number (CN).

^c Debye-Waller factor.

^d Inner potential correction.

^e Goodness of fit.

(3) Related to this is their normalization treatment to correlate the coordination number (CN) of Cu–O and Cu–Cu bonds with the Cu₂O/Cu fraction for these samples. In this method, they use the following formula $X_{\text{Cu}_2\text{O}} = \text{CN}_{\text{Cu-O}}/\text{CN}_{\text{Cu}_2\text{O}}/(\text{CN}_{\text{Cu-O}}/\text{CN}_{\text{Cu}_2\text{O}} + \text{CN}_{\text{Cu-Cu}}/\text{CN}_{\text{Cu}})$ which is a derivation of a method by Beale *et al.* for determining coordination number for mixed phases. If I understand correctly in the authors case there is no mixed phase, and thus it is simply $X_{\text{Cu}_2\text{O}} = \text{CN}_{\text{Cu-O}}/\text{CN}_{\text{Cu}_2\text{O}}$. But even here there is inconsistency. Above the authors use the justification of the increased bond length which “demonstrates a distorted tetrahedral structure owing to the interfacial Cu–O–Al coordination (CuAlO₂-like structure), and thus the normalization would be divided by the structure of CuAlO₂ and not Cu₂O.

Author reply: Thank you for this comment. According to this comment, we optimized the

computational formula, and improved the relevant content in the revised manuscript. In addition, as mentioned above, the EXAFS results give the average value, and it is difficult to distinguish the coordination information of Cu–O–Al and Cu–O–Cu based on this characterization. Nevertheless, this normalization result still has a certain significance in reflecting the difference in coordination structure of Cu–O bonds in these samples.

• **Supplementary Information, Page 4, Line 20: rephrase:**

“**Cu₂O/Cu fraction from EXAFS spectra.** A normalization treatment was performed to correlate the coordination number (CN) of Cu–O bond (or Cu–Cu) with the Cu₂O/Cu ratio for these samples. The calculation equations are as follows:

$$X_{\text{Cu}_2\text{O}} = \text{CN}_{\text{Cu-O}}/\text{CN}_{\text{Cu}_2\text{O}} \quad (9)$$

$$X_{\text{Cu}} = \text{CN}_{\text{Cu-Cu}}/\text{CN}_{\text{Cu}} \quad (10)$$

$$\gamma = X_{\text{Cu}_2\text{O}}/X_{\text{Cu}} \quad (11)$$

where the CN_{Cu₂O} and CN_{Cu} are 2 and 12, corresponding to the CNs in the first shell of Cu–O in Cu₂O and the Cu–Cu bond in Cu standard sample, respectively; the CN_{Cu–O} and CN_{Cu–Cu} denote the CNs of Cu–O and Cu–Cu bond obtained from EXAFS (Supplementary Table 2), and γ is the Cu₂O/Cu ratio for these samples.”

• Supplementary Fig. 18b has been supplemented in the revised Supplementary Information.

Supplementary Figure 18b. Coordination number of Cu–O or Cu–Cu bond as a function of Cu₂O/Cu ratio for various samples.

REVIEWER COMMENTS

Reviewer #1 (Remarks to the Author):

I have read all the changes made by the authors and they have improved the quality of their study and answered my previous concerns. Thus, I recommend acceptance for publication.

Reviewer #3 (Remarks to the Author):

While the authors have made an excellent attempt to address my concerns there are still a few points that must be addressed before I can recommend the manuscript for publication.

If we are to trust their interpretation of the validity of the difference in Cu-O bond lengths, then the error bars must be given in the main text. The bond lengths cannot be stated as 1.84 and 1.81 Å.

Even with all of the new EXAFS data it is still not clear to me if the authors tried to fit a Cu-O-Al scattering path? If they have not, then they must at least attempt this and report the results.

Supplementary Figure 20 shows the best fit to the magnitude of the FT of the Cu EXAFS data which are then enumerated in Supplementary Table 3 (according to the text in the main manuscript). However, this does not appear to be correct – the data in Table 3 only shows a fit to two scattering paths, whereas the data plotted in SI Figure 20 shows fitting to 5.5 Å.

Response to Reviewers

Reviewer #1

Comments:

(1) I have read all the changes made by the authors and they have improved the quality of their study and answered my previous concerns. Thus, I recommend acceptance for publication.

Author reply: Thank you for this comment.

Reviewer #3

Comments:

While the authors have made an excellent attempt to address my concerns there are still a few points that must be addressed before I can recommend the manuscript for publication.

(1) If we are to trust their interpretation of the validity of the difference in Cu-O bond lengths, then the error bar must be given in the main text. The bond lengths cannot be stated as 1.84 and 1.81 Å.

Author reply: Thank you for this comment. According to this comment, we have added the error bar for the Cu–O bond length in the revised manuscript.

• **Page 8, Line 11: rephrase:** “... the longer bond length of Cu–O in γ Cu/Cu(Al)O_x samples (1.84±0.01 Å) relative to Cu₂O standard (1.81±0.01 Å) indicates a distorted tetrahedral structure due to the partial substitution of Cu by Al.”

(2) Even with all of the new EXAFS data it is still not clear to me if the authors tried to fit a Cu-O-Al scattering path? If they have not, then they must at least attempt this and report the results.

Author reply: Thank you for this comment. According to this comment, we performed fitting on the Cu–O–Al scattering path with CIF data of CuAlO₂ as the standard model. The corresponding results

and discussion have been added in the revised manuscript. Supplementary Fig. 20 and Supplementary Table S3 have been modified in the revised Supplementary Information.

• **Page 9, Line 12: rephrase:** “Meanwhile, an enhanced proportion of Cu–O–Al geometric coordination is further demonstrated through fitting the Cu–O and Cu–O–Al scattering paths based on Cu K-edge EXAFS spectra with Cu₂O and CuAlO₂ as standard models (Supplementary Figs. 19f, 20 and Supplementary Table 3). The average bond length of Cu–O increases from 1.84±0.01 (4.25Cu/Cu(Al)O_x-500) to 1.89±0.02 Å (4.25Cu/Cu(Al)O_x-800) in sequence, accompanied with a gradual increase in the coordination number of Cu–Al bond from the second shell (bond length: ~3.17 Å), which indicates the formation of Cu–O–Al geometric coordination.”

• Supplementary Fig. 20 and Supplementary Table 3 have been improved in the revised Supplementary Information.

Supplementary Figure 20. Fitting results of EXAFS spectra at the Cu K-edge of **a** 4.25Cu/Cu(Al)O_x-500, **b** 4.25Cu/Cu(Al)O_x-600, **c** 4.25Cu/Cu(Al)O_x-700 and **d** 4.25Cu/Cu(Al)O_x-

800 samples (black line: the experimental data; red line: the fitting curve; blue line: the residual data).

Supplementary Table 3. EXAFS fitting parameters at the Cu K-edge for various samples

Sample	Shell	R (Å) ^a	CN ^b	σ^2 (10^{-3}Å^2) ^c	ΔE_0 (eV) ^d	R factor (%) ^e
Cu-foil	Cu–Cu	2.54(0.01)	12.0(0.2)	8.5	4.52	0.2
Cu₂O	Cu–O	1.81(0.01)	2.0(0.1)	3.7	8.17	2.0
	Cu–Cu	3.01(0.02)	12.0(0.1)	3.4		
4.25Cu/Cu(Al)O_x-500	Cu–O	1.84(0.01)	0.9(0.2)	9.1	5.96	1.1
	Cu–Cu	2.54(0.01)	6.9(0.2)	9.1		
	Cu–Al	3.17(0.03)	1.7(0.1)	9.1		
4.25Cu/Cu(Al)O_x-600	Cu–O	1.85(0.01)	1.1(0.2)	9.0	3.99	1.5
	Cu–Cu	2.54(0.01)	6.3(0.1)	9.0		
	Cu–Al	3.18(0.02)	2.1(0.2)	9.0		
4.25Cu/Cu(Al)O_x-700	Cu–O	1.87(0.02)	1.4(0.2)	8.9	7.64	1.2
	Cu–Cu	2.54(0.01)	5.6(0.3)	8.9		
	Cu–Al	3.18(0.03)	2.5(0.1)	8.9		
4.25Cu/Cu(Al)O_x-800	Cu–O	1.89(0.02)	1.7(0.2)	9.6	4.27	1.5
	Cu–Cu	2.55(0.02)	4.9(0.3)	9.6		
	Cu–Al	3.18(0.04)	2.8(0.2)	9.6		

^a Bond length.

^b Coordination number (CN).

^c Debye-Waller factor.

^d Inner potential correction.

^e Goodness of fit.

(3) Supplementary Figure 20 shows the best fit to the magnitude of the FT of the Cu EXAFS data which are then enumerated in Supplementary Table 3 (according to the text in the main

manuscript). However, this does not appear to be correct – the data in Table 3 only shows a fit to two scattering paths, whereas the data plotted in SI Figure 20 shows fitting to 5.5 Å.

Author reply: Thank you for this comment. We are sorry for this description mistake. In Supplementary Fig. 20, the black line represents the experimental data; the red line represents the fitted data (1–3 Å), and the blue line denotes the residual unfitted data (3–6 Å). In the original manuscript, we only enumerated the fitted scattering paths (red line) in Supplementary Table 3. According to this comment, in order to avoid misunderstanding, we have added the relevant explanation in the caption of Supplementary Fig. 20. In addition, we performed fitting on the Cu–Cu, Cu–O and Cu–Al bonds with Cu foil, Cu₂O and CuAlO₂ as the standard models, and the data fitting ranges for k-range, R-range and S02 value are 2.5–12.0 Å⁻¹, 1.0–3.5 Å and 0.8, respectively. Supplementary Fig. 20 and Supplementary Table S3 have been improved as shown in question (2).

REVIEWERS' COMMENTS

Reviewer #3 (Remarks to the Author):

The authors have addressed my comments to my satisfaction. The quality of the manuscript has been improved substantially. Thus, I recommend acceptance for publication.

Response to Reviewers

Reviewer #3

Comments:

The authors have addressed my comments to my satisfaction. The quality of the manuscript has been improved substantially. Thus, I recommend acceptance for publication.

Author reply: Thank you for this comment.